# From calibration to parameter learning: Harnessing the scaling effects of big data in geoscientific modeling

Wen-Ping Tsai [1], Dapeng Feng[1], Ming Pan [2,3], Hylke Beck [4], Kathryn Lawson [1,5], Yuan Yang [6,7], Jiangtao Liu[1] & Chaopeng Shen [1,5 ✉]

The behaviors and skills of models in many geosciences (e.g., hydrology and ecosystem sciences) strongly depend on spatially-varying parameters that need calibration. A well-calibrated model can reasonably propagate information from observations to unobserved variables via model physics, but traditional calibration is highly inefficient and results in non-unique solutions. Here we propose a novel differentiable parameter learning (dPL) framework that efficiently learns a global mapping between inputs (and optionally responses) and parameters. Crucially, dPL exhibits beneficial scaling curves not previously demonstrated to geoscientists: as training data increases, dPL achieves better performance, more physical coherence, and better generalizability (across space and uncalibrated variables), all with orders-of-magnitude lower computational cost. We demonstrate examples that learned from soil moisture and streamflow, where dPL drastically outperformed existing evolutionary and regionalization methods, or required only ~12.5% of the training data to achieve similar performance. The generic scheme promotes the integration of deep learning and process-based models, without mandating reimplementation.

[1] Civil and Environmental Engineering, Pennsylvania State University, University Park, PA, USA. [2] Center for Western Weather and Water Extremes, Scripps Institution of Oceanography, University of California San Diego, La Jolla, CA, USA. [3] Civil and Environmental Engineering, Princeton University, Princeton, NJ, USA. [4] GloH2O, Almere, the Netherlands. [5] HydroSapient, Inc, State College, PA, USA. [6] Department of Hydraulic Engineering, Tsinghua University, Beijing, China. [7] Institute of Science and Technology, China Three Gorges Corporation, Beijing, China. ✉email: cshen@engr.psu.edu

This work broadly addresses geoscientific models across a wide variety of domains, including non-dynamical system models like radiative transfer modeling[1], as well as dynamical system models such as land models that are used in Earth System Models; hydrologic models that simulate soil moisture, evapotranspiration, runoff, and groundwater recharge[2]; ecosystem models that simulate vegetation growth and carbon and nutrient cycling[3]; agricultural models that simulate crop growth[4]; and models of water quality[5] and human-flood interactions[6]. Besides scientific pursuits, these models fill the operational information needs for water supply management, pollution control, crop and forest management, climate change impact estimation, and many others.

A central and persistent problem concerning this wide variety of geoscientific models is that their behaviors and skills are strongly impacted by unobservable and underdetermined parameters. Fundamentally, a geoscientific model can be regarded as a mapping function from some time- and location-specific inputs (**x**) to some time- and location-specific outputs (**y**), modulated by characteristics of that location (observable attributes **A** and unobserved parameters θ) (see Methods for a more formal, mathematical description). A calibration algorithm seeks to adjust the values of the unobserved parameters (θ) at each location, so that the difference between the model's outputs (**y**) and some independent measurements (**z**) is minimized. But uncertainties in these parameters, such as for those controlling the simulated land surface feedbacks of water and $CO_2$ to the atmosphere, limit the confidence we have in the modeled results, such as simulated regional impacts caused by increasing $CO_2$ levels[7].

For decades, parameter calibration has been the orthodox procedure that is deeply entrenched across various geoscientific domains. An entire research enterprise and many textbook chapters have been dedicated to these calibration techniques and their applications in geosciences. Myriad methods including genetic algorithms[8–10] and evolutionary algorithms (EAs) such as the shuffled complex evolution method (SCE-UA)[11] have been introduced for calibration. For example, nearly all models for the rainfall-runoff process[12,13] and for ecosystem dynamics[14] involve unobservable parameters that require calibration. Moreover, these parameters are often sensitive to changes in spatial and/or temporal resolution[15], other model parameters, model version, and input data, continuously triggering the need to readjust previously calibrated parameters – a repetitive and tedious process[16]. Current optimization algorithms require thousands of model runs or more, just to calibrate a dozen parameters.

Geoscientific processes have commonalities and dissimilarities between regions which could potentially be collectively leveraged by a big-data learning procedure. However, because traditional calibration procedures are generally applied to each location individually, they cannot exploit such commonalities: in other words, they do not take advantage of what is learned in one place to apply it elsewhere. Because of the small amount of data at each site, algorithms may overfit to training data and find non-physical parameters, meaning they captured noise instead of a true signal. This often leads to the dreaded non-uniqueness problem (i.e., equifinality)[17–19], where wildly different parameter sets produce similar evaluation metrics and thus cannot be determined by calibration. Site-by-site calibration often produces disparate, discontinuous parameters for neighboring, geographically-similar areas, as shown for hydrologic models[16]. In summary, the traditional parameter calibration paradigm has become a bottleneck and a distinctive pain point to large-domain modeling in geoscientific research.

A class of techniques collectively referred to as parameter regionalization attempt to apply a more stringent constraint using all available data, which can help alleviate these issues[20]. A specific type of regionalization scheme prescribes transfer functions to relate known physical attributes to parameters[15,21]. The structures of the transfer functions are determined by humans (and thus need to be specifically customized for each model and data source), and the rigid form often limits the efficacy in predicting parameters. As we will show, known regionalization schemes generate sub-optimal solutions that are fundamentally not ready to leverage big data. Additionally, they cannot handle a large number of parameters in the transfer functions, and are restricted to simple input attributes.

Recently, deep learning (DL)[22,23], a category of neural networks with multiple layers and specialized architecture to learn patterns from spatial or temporal structures of data, has shown great promise across scientific disciplines, including the geosciences[24–26], although some limitations also emerged. In the field of hydrology, previous work including ours has shown that time series DL network models based on algorithms such as the long short-term memory (LSTM) algorithm[27] have had success predicting soil moisture[28–30], streamflow[31], stream temperature[32], dissolved oxygen[33], and lake water temperature[34]. However, such a data-driven modeling method only allows for predicting observable variables for which we have sufficient data. For similarly important but unobserved prognostic variables such as evapotranspiration, groundwater recharge, or root carbon storage, we still rely on manually-calibrated process-based models (PBM).

Modern DL networks and their highly efficient training procedures, i.e., backpropagation and gradient descent, are well-suited to exploit the information in large datasets. One could envision leveraging DL to solve parameter calibration at large scales, but despite repeated calls to integrate physics and DL methods[24], to the best of our knowledge there are no frameworks that exploit modern DL for the parameter calibration problem. More specifically, backpropagation needs differentiable computing: we need to analytically track the derivative of outputs with respect to the inputs for every calculation step in the model. Most modern machine learning platforms support automatic differentiation (AD) which automatically keeps track of all gradients, but traditional programming environments do not, and reimplementing existing models on differentiable platforms would incur huge costs for expert labor. Without AD, derivatives may also in theory be approximated by finite difference, but this is less accurate and computationally intractable for large neural networks. While neural networks have been used in traditional parameter calibration, their typical, shallow role has been that of an efficient surrogate model, which emulates a PBM to reduce computational time during calibration[35]. With that paradigm, the calibration problems are still solved independently for each site. We propose a much deeper integration between DL and PBM.

Here we propose a differentiable parameter learning (dPL) framework based on deep neural networks, with two versions ($g_A$ and $g_z$) suitable for different use cases in geosciences. The overall dPL framework contains a parameter estimation module (Fig. 1b, c, equations in Methods section) that maps from raw input information (either observable attributes **A** alone for $g_A$, or forcing-response pairs **x**-**z** for $g_z$) to PBM parameters, which are then fed into a differentiable PBM (or, alternatively, a surrogate model like a neural network. See Fig. 1a). On the programming level, a PBM is differentiable if it is compatible with and directly implemented on an AD-supported DL platform like PyTorch[36], Tensorflow[37], or Julia[38], while modern neural networks are already differentiable. In essence, we learn the mapping relationship $g_A$ or $g_Z$ from the inputs of **A** or (**A**, **x**, **z**), to the optimal parameters (Fig. 1c). This method can then use model physics to propagate information from observations to unobserved variables.

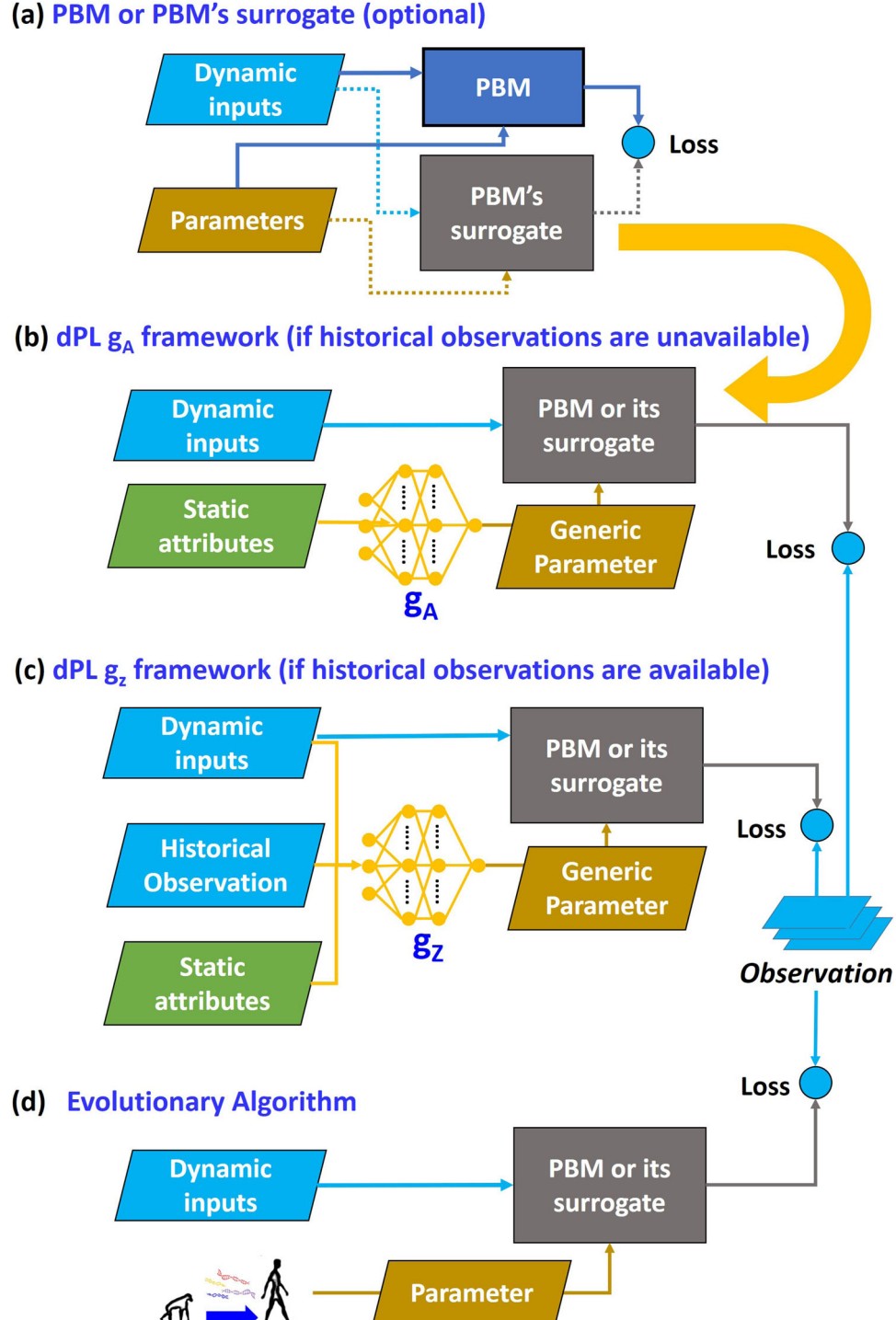

**Fig. 1 Comparison of the dPL framework to the traditional calibration paradigm. a** A deep learning model is trained to mimic the outputs of a process-based model (PBM). This step is optional since one may also directly implement the model in a DL platform. **b** Workflow of the first dPL option, network $g_A$: parameters are inferred by a network (in our case, a separate LSTM network) based on auxiliary attributes. These parameters are then sent into the PBM, whose outputs are compared to the observations to calculate the loss (the difference between objective function and observation). **c** Workflow of the second dPL option, network $g_Z$: historical observations (meteorological forcings and observed responses) are additional inputs to the parameter estimation network. **d** Traditional site-by-site parameter calibration framework.

There are four salient features with this approach. First, the loss function is defined over the entire training dataset to train such a mapping (a global constraint), unlike the traditional location-specific objective function. Second, the use of differentiable computing and gradient-based optimization supports learning a complex mapping with a global constraint. Third, we employ an end-to-end framework in the sense that the training targets are observed variables like soil moisture or streamflow, rather than intermediate variables such as parameters estimated by other means. Our framework transforms the typically inverse parameter calibration problem into a big-data DL problem, leveraging the efficiency and performance of the modern DL computing

infrastructure. Fourth, (for $g_Z$ only) we learn the mapping from forcing and response pairs (**x**-**z**) to parameters.

In the first and main case study of this work, we apply the dPL framework to the widely used Variable Infiltration Capacity (VIC) land surface hydrologic model[39], producing parameters which allow VIC to best match surface soil moisture observations from NASA's Soil Moisture Active Passive (SMAP) satellite mission[40]. We compare parameters from dPL to those from the standard evolutionary algorithm SCE-UA, and also to the operational parameters from the widely used North American Land Data Assimilation System phase-II dataset (NLDAS-2)[41]. The comparisons are done at multiple training sampling densities with different lengths of training data. We evaluate the quality of the estimated parameters for locations outside of the training set, and also for an uncalibrated variable (evapotranspiration, ET). In the second case study, we train the framework on the CAMELS dataset[42] which consists of 531 basins in the US, with daily streamflow as the target and VIC as the PBM. We compare our results to the Multiscale Parameter Regionalization (MPR) approach[21]. In the third case study, we train on a global streamflow dataset for spatial extrapolation, or prediction in ungauged basins (PUB), in comparison to another state-of-the-art regionalization scheme recently published by Beck et al.[43]. In the first two cases we train a separate neural network as a differentiable surrogate for VIC, and in the third case we directly implement the simple conceptual hydrologic model Hydrologiska Byråns Vattenbalansavdelning (HBV) in a DL platform.

## Results

**Optimization performance and efficiency.** For the SMAP calibration case study, our results (Fig. 2) show that dPL can deliver equivalent or slightly better ending performance metrics than the evolutionary algorithm SCE-UA over the entire contiguous United States (CONUS). At the moderate training sampling density of $1/4^2$, where $1/4^2$ represents sampling one gridcell from each 4x4 patch (also abbreviated as s4, see Fig. S1 in Supplementary Information for illustration), dPL had nearly identical ending error metrics (root-mean-square error, RMSE, between the simulated and observed surface soil moisture) as SCE-UA (Fig. 2a, b). dPL's marginal outperformance (or virtual equivalence) at $1/4^2$ was a surprise to us, as one would expect an EA like SCE-UA to best capture the global minimum. This result attests to the uncompromising optimization capability of gradient descent. It also suggests there are commonalities to be leveraged in hydrologic processes across different sites. We observe that dPL's performance is related to the amount of training data: it had the lowest performance (highest ending RMSE) when there was only 1 year's worth of training data with the lowest sampling density ($1/16^2$), and the highest performance when there was 2 years' worth of training data with the highest sampling density ($1/4^2$).

Notably, as training data amount increased, dPL descended into the range of acceptable performance orders of magnitude faster than SCE-UA in terms of both the number of forward runs (Fig. 2a, b) and computational time (Fig. 2c, d) per gridcell. For the model trained for 2 years, dPL required 810, 370, and 45 full forward runs per gridcell (or 2.2, 0.74, and 0.31 seconds of computing time per gridcell, not proportional to forward runs due to increasing hidden sizes) to drop below the threshold for a functional model (RMSE = 0.05) at $1/16^2$, $1/8^2$, and $1/4^2$ sampling densities, respectively. In contrast, SCE-UA needed 950 runs per gridcell (or 90 seconds, here we did not implement parallelism for SCE-UA) to reach the same RMSE with the same surrogate model running on GPU, which was similar between sampling densities $1/16^2$, $1/8^2$, and $1/4^2$. Two factors are at play for dPL's efficiency: the first factor is the reduction of runs at

higher sampling densities (with an order of magnitude spread between $1/16^2$ and $1/4^2$ in terms of either runs or time). This super scaling effect resulted from the use of a domain-wide loss function and mini-batch training, allowing dPL to learn across locations rapidly (more interpretation in Discussion). The second factor is the inherent gridcell-level (minibatch) parallelism and efficient GPU computing, which are crucial to the success of DL[22] and were made relevant to parameter estimation via dPL. While SCE-UA can also be parallelized, it may be difficult to achieve the high level of parallel efficiency and scale enjoyed by dPL.

While using a surrogate is not novel, the LSTM surrogate model enabled the differentiable computing workflow and further saved an order of magnitude of computational time as compared to the VIC model running on CPU. Strikingly, using the same criterion (RMSE = 0.05), it takes dPL roughly 25 min at $1/4^2$ sampling density on a single GPU (NVIDIA 1080Ti with 11GB memory) while it would take 33 days for a single CPU, or 475 min for a 100-core CPU cluster assuming perfect parallelism. Training the surrogate model (see Methods section) also required multiple iterations of CONUS-scale forward simulations. All things considered, the new dPL framework brings a difference of 2–3 orders of magnitude in time, not to mention the savings in energy. This is despite the fact that dPL trains large neural networks with thousands of weights. While there are more efficient variants of SCE-UA, we compared dPL to the standard algorithm because it is well understood and benchmarked, and its variants do not differ in efficiency by orders of magnitude.

**Spatial extrapolation and uncalibrated variables.** Compared to SCE-UA, dPL generalized better in space and gave parameter sets that were spatially coherent and better constrained, especially as the amount of data increased. Our spatial generalization test showed dPL's metrics had almost no degradation from the training set to the out-of-training neighboring gridcells (Fig. 3d). In contrast, SCE-UA's ubRMSE increased for the neighboring gridcells, with a statistically significant difference ($p = 0$ based on Wilcoxon signed-rank test). A similar pattern is observed in another year (Supplementary Fig. S5). More apparently, dPL had a much smaller spread of bias compared to SCE-UA (Fig. 3d left panel). $g_Z$ was slightly better than $g_A$, and both were better than SCE-UA in the spatial generalization test.

These comparisons suggest that dPL learned a more robust parameter mapping pattern than SCE-UA, a strength we attribute to using the global constraint. The difference in metrics between SCE-UA and dPL was statistically significant and random seeds can be rejected as a cause (Table S1 in Supplementary Information). However, the difference may not appear large, which is to be expected as the difference was bounded by soil moisture physics, similar atmospheric inputs, and spatial proximity to training sites. A larger difference is noted in the streamflow examples below.

Most geoscientific models output multiple unobserved variables of interest that can be used as diagnostics or to support narratives of the simulations. It can be argued that if our parameter set leads to improved behavior for both calibrated (soil moisture) and uncalibrated (evapotranspiration, ET) variables, it better describes the underlying physical processes, and the model gives good results for the right reason. Here we compared model-simulated temporal-mean ET to MOD16A2, an ET product from the completely independent Moderate Resolution Imaging Spectroradiometer (MODIS) satellite mission (see Methods section for discussion of limitations).

The parameters from dPL produced ET that was closer to MOD16A2 in spatial pattern than did either the parameters from NLDAS-2 or those calibrated by SCE-UA (Fig. 4). At $1/8^2$

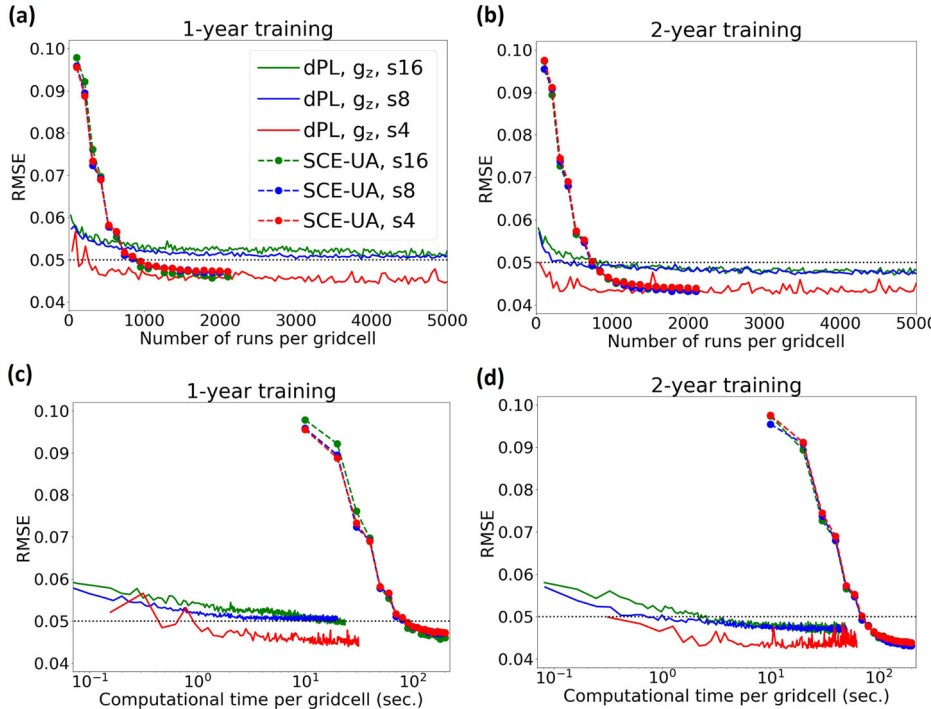

**Fig. 2 Objective function (root-mean-square error, RMSE) and computational time for the testing period vs. number of forward runs per gridcell.** Here, a forward run for dPL means running through the same number of days as the specified training period, e.g., 30 random instances of 240-day-long simulations in a minibatch would be counted as ~20 runs for a 2-year training period. Dashed lines are for SCE-UA and solid lines are for dPL. s16, s8, and s4 denote models trained with sampling densities of 1/16², 1/8², and 1/4², respectively, where 1/16² represents sampling one gridcell from each 16 × 16 patch. The dotted horizontal line represents the RMSE threshold of 0.05, below which a model is considered functional. **a** RMSE for the models trained with 1 year's worth of data. **b** Same as **a** but for models trained with 2 years' worth of data. **c** RMSE vs computational time per gridcell. Both methods use the same surrogate model running on one GPU, and 1 year's worth of training data. **d** Same as **c** but for models trained with 2 years' worth of data.

sampling density, the CONUS-median (ensemble mean) values for correlation between observed and simulated ET were 0.75 and 0.69 for $g_Z$ and SCE-UA respectively, and the differences were multiple standard deviations (due to random seeds) apart (Table S1 in Supplementary Information). Due to the much smaller bias, the Nash-Sutcliffe model efficiency coefficient (NSE) for dPL ($g_Z$) was 0.55, as opposed to 0.38 for SCE-UA and 0.44 for NLDAS-2. SCE-UA calibration using soil moisture did not improve the spatial pattern of ET compared to the current parameter sets in NLDAS-2, but dPL led both by a substantial margin. Similar to soil moisture, ET variation is strongly driven by rainfall and energy inputs, so we should not expect the model to give wildly worse results even if parameter sets are not ideal.

While MOD16A2 should not be considered truth, it utilizes completely separate sources of observations including leaf area index and photosynthetically active radiation (see Methods section). Thus, the better agreement of dPL with MOD16A2, in terms of both correlation and bias, suggest that dPL had more physically-relevant parameter sets. When SCE-UA calibrated a certain gridcell, it did not put this gridcell in the context of regional patterns, so it could distort model physics in its pursuit of lowest RMSE in soil moisture for that location. For dPL, because the inputs to the parameter estimation module, i.e., forcings, responses, and attributes, are themselves spatially coherent (autocorrelated), and only one uniform model is trained, the inferred parameters are also spatially coherent.

The inferred parameter fields reveal the reason behind the advantage of dPL over SCE-UA. One of the parameters estimated by dPL, INFILT (see Methods section), showed a spatial pattern that generally follows the aridity and topographic patterns of the CONUS (Fig. 5a), which agrees with our general understanding of

physical hydrology and the VIC model's behavior. Precipitation declines from east to west until reaching the Rocky Mountains, and then increases again at the west coast. The large southeast and northwest coasts are the wettest parts of the country due to moisture from the oceans. dPL kept the steepness of the infiltration capacity curve of surface runoff smooth in wet areas to produce more runoff, which is consistent with earlier literature[16,44]. INFILT varied continuously in the southeast coastal plains where soil is thick and permeable and most rainfall infiltrates[45]. dPL also captured patterns of poorly drained soils at the northeastern edge of the map, which are also visible from soil surveys. The spatial autocorrelation was high from the beginning to the end of the optimization (Fig. 6a), showing the global constraint allows the model to learn across sites from the beginning. We observed such continuity with other parameters as well (Supplementary Fig. S3).

In contrast, SCE-UA (Fig. 5b) presented discontinuous parameters apparently plagued by stochasticity and parameter non-uniqueness, which explains why SCE-UA had worse performance in the spatial generalization and uncalibrated variable tests. Soil moisture observations impose constraints only on a part of the system, and VIC, like any PBM, also contains structural deficiencies; therefore, it is unreasonable to expect dPL (or SCE-UA, or any other scheme) to fully remove parametric uncertainties or find the most realistic parameters. Nevertheless, the parameters found by dPL seem more coherent with known physical relationships than those from SCE-UA.

**Streamflow cases in comparison with regionalization schemes.** The streamflow temporal generalization scenario showed even

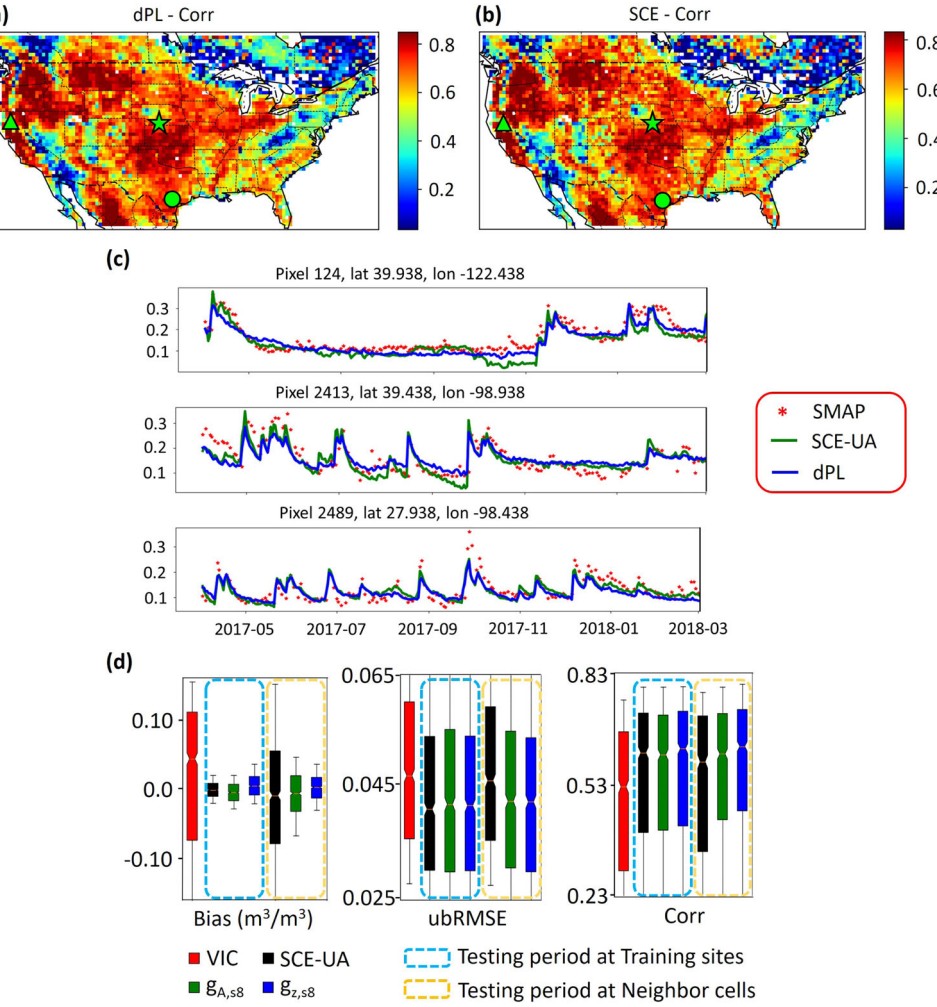

**Fig. 3 Performance of SMAP soil moisture simulations. a** Map of correlations between observed (Soil Moisture Active Passive satellite, SMAP) and simulated (dPL $g_Z$) soil moisture during test period at $1/4^2$ sampling density. Green symbols (triangle, star, and circle) indicate three randomly selected sites. **b** Same as **a** except using simulated soil moisture from evolutionary algorithm SCE-UA. **c** Example time series at three randomly selected sites. **d** Boxplots summarizing gridcell metrics for temporal generalization (evaluated on the training locations over the test period, blue dashed box) and spatial generalization (orange dashed box) tests at $1/8^2$ sampling density (one gridcell from an 8x8 patch, see illustration in Supplementary Fig. S1) for VIC (NLDAS-2 default parameters), SCE-UA, and dPL options $g_Z$ and $g_A$. In the spatial generalization test, we sampled at $1/8^2$ density for training and evaluated the parameters' performance on a neighbor 3 rows to the north and 3 columns to the east from each of the training gridcells, over the test period. Results from another year are similar (Supplementary Fig. S5). We tested on other neighboring gridcells as well, with similar results (data not shown). The boxplots' lower whisker, lower box edge, middle line, upper box edge, and upper whisker represent 10th, 25th, 50th, 75th, and 90th percentiles, respectively.

more pronounced advantages of dPL over existing state-of-the-art method multiscale parameter regionalization (MPR). In the case with the CAMELS hydrologic dataset (Supplementary Fig. S4a), we applied dPL to estimate parameters for the VIC hydrologic model and tested the parameters in a different period of time (temporal generalization) on basins in the training set. $g_Z$ achieved a median NSE of ~0.44, compared to the median value of 0.32 reported for MPR[21] (Fig. 6a). This result challenges the previous argument that a low value of 0.32 in this experimental setup was close to the performance ceiling of VIC due to its structural deficiencies, and showed that the regionalization scheme was also not near optimal. The dPL results seemed to be worse than MPR where NSE was <0.2, which we think is due to the lower quality of the surrogate model on those difficult-to-simulate basins. However, in this NSE range, the models are already known to be unreliable.

Despite the improved parameter learning scheme, the best results achievable with VIC (median NSE = 0.44) are much

weaker than a pure LSTM model. LSTM can obtain a median NSE of 0.74 (with the use of an ensemble)[31], which may be interpreted as being close to the best possible model given forcing and attribute errors. The gap from 0.44 to 0.74 can be partially explained by the imperfectness of the surrogate model, but it is unlikely to be the major culprit based on its agreement with VIC (Supplementary Fig. S2). Given dPL's strong optimizing capability, the remaining gap can be mostly attributed to the limited understanding of hydrology as encoded in the structures of VIC or HBV. There is potential in the future to extend our framework to further learn better model structure to narrow the gap, and, by doing so, improve our understanding of the physical system.

A large advantage of dPL was also noticed in the global PUB case (spatial extrapolation), where we applied dPL with two sets of input features on a global hydrologic dataset (Supplementary Fig. S4b) and tested on basins not included in the training set. An existing state-of-the-art regionalization scheme from Beck et al.[43] (hereafter referred to as Beck20) reported a median Kling-Gupta

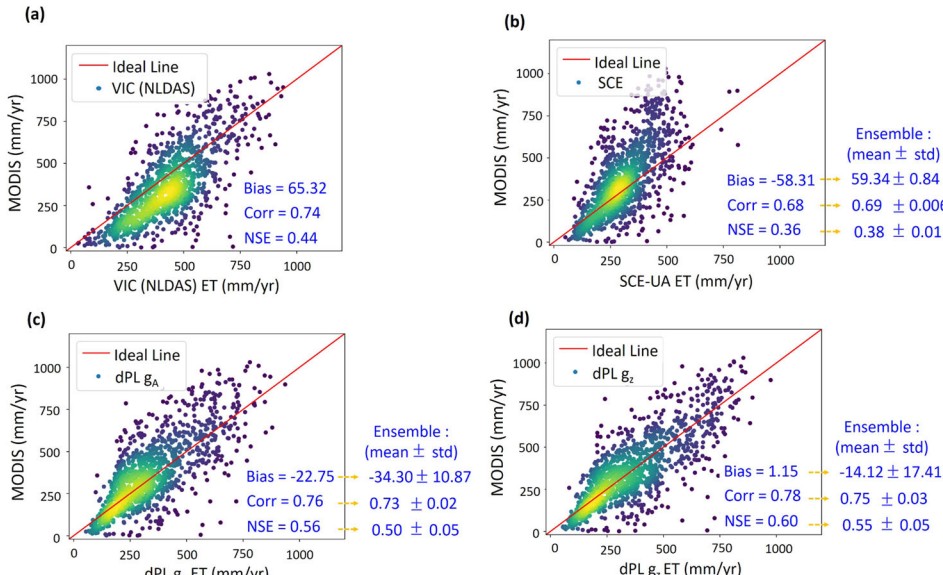

**Fig. 4 Uncalibrated variable (evapotranspiration, ET) metrics from models trained at 1/8² sampling density.** Scatter plots of temporal-mean ET (mm/year) comparing the MOD16A2 satellite product with ET produced by **a** NLDAS-2, **b** SCE-UA, and dPL options **c** $g_A$, and **d** $g_Z$. Each point on the scatter plot is the temporal mean ET of a 1/8-latitude-longitude-degree gridcell defined on the NLDAS-2 model grid. Yellow color indicates higher density of points. The ensemble metrics are from training the model with different random seeds, while the 1-vs-1 plots came from one particular random realization. Panel **a** is for the NLDAS-2's default VIC model simulation and does not have an ensemble.

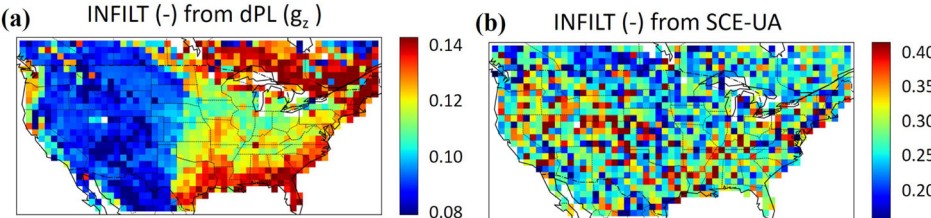

**Fig. 5 Comparison of parameters generated by dPL and SCE-UA.** The continuous, spatially representative patterns of **a** dPL-inferred parameters are noteworthy, especially in comparison to the discontinuous, random appearance of **b** SCE-UA-inferred parameters from site-by-site calibration. Both were trained with a 1/8² sampling density.

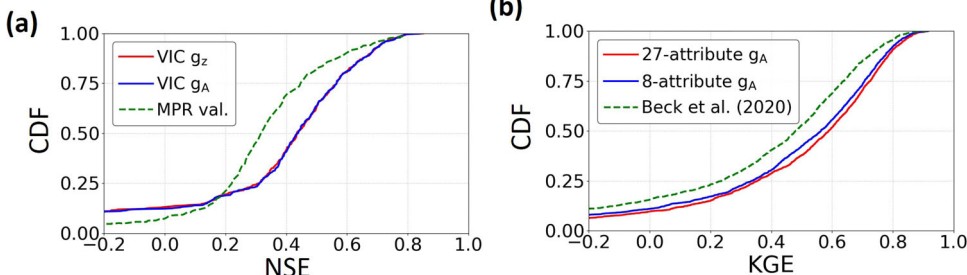

**Fig. 6 Comparison of dPL and regionalization schemes for streamflow calibration. a** Calibrating the VIC hydrologic model via a differentiable surrogate model on the CAMELS dataset over the conterminous United States, in comparison to Multiscale Parameter Regionalization (MPR). **b** Calibrating the HBV hydrologic model (not a surrogate) on the Beck20 global dataset, in comparison to the Beck20 regionalization scheme. We used NSE for the CAMELS case and KGE for the global case because these metrics were used by the respective papers, and the main purpose of the case studies was to compare with the existing literature. For both panels, curves on the right represent better models.

efficiency coefficient (KGE, similar in magnitude to NSE, see Methods section) of 0.48 for the temperate catchment group[43], while dPL gave values of 0.56 for the comparable 8-feature setup (Fig. 6b). We witnessed a noticeable separation of the cumulative distribution function (CDF) curves between dPL and Beck20 throughout the different ranges of KGE. In addition, the 27-feature dPL setup generalized better in space than did the 8-feature dPL setup (median KGE = 0.59), suggesting that using more attributes as inputs did not cause dPL to overfit. It would take considerably more effort for the traditional schemes to run the 27-feature setup as it entails including more transfer functions and more parameters to train.

These differences of around 0.1 in median NSE or KGE in both of temporal and spatial experiments are quite significant, as NSE = 1 indicates a perfect model while NSE = 0 corresponds to using the mean value as the prediction. Increases in NSE from 0.32 to 0.55 in the VIC case, or from 0.48 to 0.56 in the HBV case represent consequential changes in model reliability for water management planning applications. Again, for all cases, the dPL metrics are still considerably lower than what would be obtained by purely data-driven LSTMs. Rather than being caused by the dPL algorithm, this difference is more likely due to the limitations of the PBM structures and the ways the reference problems were set up in the literature, e.g., choices of inputs, assuming homogeneity at the basin level, no routing in the global PUB case, etc. We inherited these setups because the main purpose here is to compare with previous schemes. While traditional regionalization schemes like MPR are an important and constructive avenue (e.g., they can also generate spatially-smooth parameter fields looking like Fig. 5a), our comparisons suggest that they are far from optimal and thus cannot fully leverage information provided by big data. The sub-optimality is because (i) the transfer functions were human expert-derived and cannot extract as much useful information contained in inputs as a deep network; and (ii) their optimization schemes have limitations in handling of large datasets and many parameters as problem complexity grows.

dPL offers the flexibility to leverage all forms of available information. It is not possible for regionalization schemes like MPR to map from time series responses ($z$) to parameters as $g_Z$ does. For soil moisture, our tests showed that enabling learning from $x$-$z$ pairs with $g_Z$ improved the simulations with statistical significance compared to using the attributes alone with $g_A$ for soil moisture, evapotranspiration, and neighboring gridcell soil moisture (Table S1 in Supplementary Information). The information contained in $z$ seems to have improved the physical significance of the parameters.

**Scaling behavior with respect to the amount of training data (scaling curves).** Summarizing the results in another way, as the amount of training data increases, we clearly witness beneficial scaling behaviors (Fig. 7a), which to our knowledge have never been discussed in the context of geoscientific modeling. As training data increases, the performance (based on the ending RMSE for soil moisture), physical coherence (based on the

uncalibrated variable ET), and generalization (based on spatial extrapolation) all improve, while the average cost per site decreases dramatically (based on number of forward runs). The reduction in cost can be interpreted as dPL demonstrating economies of scale (EoS), where a mildly rising global training cost is shared by all sites. But beyond EoS, the improvements in parameter performance and physical coherence indicate that each site now benefits from a better "service" resulting from the participation of other sites. Each additional data point allows the data-driven scheme to better capture details of the underlying parameter-response function, and more data imposes a stronger large-scale physiographical constraint that must be simultaneously satisfied, suppressing overfitting and improving robustness. This scaling effect is an important reason why dPL can surpass SCE-UA: at the low data-density end (s16 and s8 in Fig. 2), dPL's ending RMSE in fact was not as strong as that of SCE-UA.

For the global PUB experiments, we witnessed a two-phase scaling curve and strong data efficiency with dPL when training data was systematically reduced (Fig. 7b). In the first phase (2–25% of basins used in training), there was a rapid rise where the median KGE improved from 0.38 to 0.54. In the second phase, the improvement became much slower but did not saturate at 100% of the dataset. There is an upper bound to the PBM performance due to model structural deficiencies, so the slowdown is to be expected, but the initial scaling curve is surprisingly steep. We suspect the first rapid-rise phase is due to dPL's ability to learn across sites (at 25% basin density, the model mostly learned the main characteristics of the problem), while the second, more asymptotic phase is due to reduced geographic distance between training and test basins (the model does fine tuning).

Overall, if we assume the Beck20 scheme to also improve monotonously with more basins on the PUB experiment (this is likely as more training basins mean smaller distances from test basins, akin to the second phase described above), dPL would achieve the same performance as Beck20 (median ~KGE = 0.48) using just ~12.5% of the training data. It would be interesting to compare the dPL scaling curve to that of Beck20 but, unfortunately due to prohibitive computational and time costs, we could not run the same experiments for Beck20. Nevertheless, it should be fair to say dPL has a higher upper bound in performance and better data efficiency.

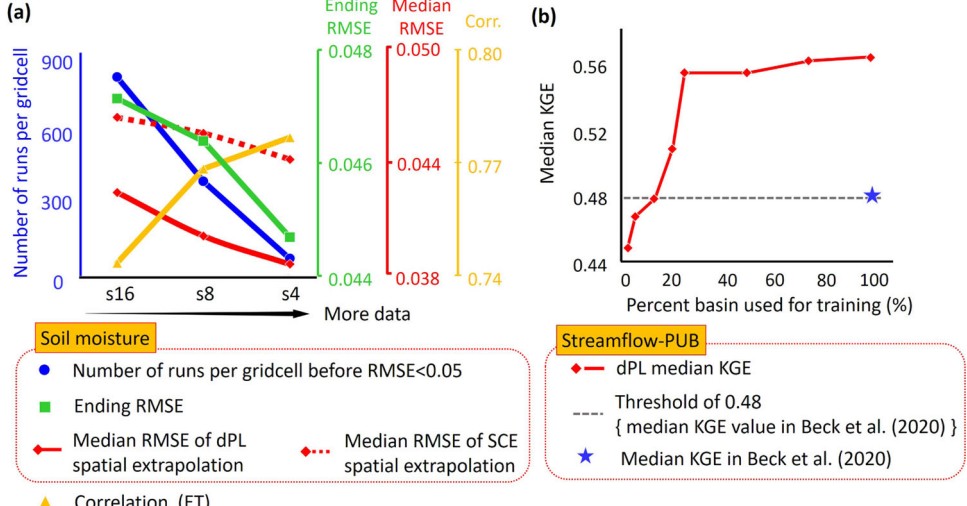

**Fig. 7 Scaling curves of dPL. a** Training data sampling density increases from s16 to s4. Dashed and solid red curves share the same red *y* axis (dashed line is for SCE-UA, to enable comparison). **b** Scaling curve for the spatial extrapolation (PUB) test with the Beck20 global headwater catchment dataset.

## Discussion

There are several major implications of the results. First, the novel scaling curves, which we expect to hold true for other geoscientific domains, showed that dPL's advantages largely arise from leveraging big data and the process commonalities and differences found therein. Without big data, such advantage may not exist or can shrink substantially. The simultaneous benefits of data scaling with respect to performance, efficiency, and (unique to parameter learning) physical coherence had not yet been demonstrated in geosciences and some other domains. These results give geoscientists strong motivation to rise above case-specific datasets (which is currently a common practice even among machine learning studies, based on our review[25]), compile large datasets, and collectively reap the benefits of such scaling curves. The curves also suggest that any interpretation of DL model results must be grounded in the context of training data amount; e.g., comparisons involving DL in a small-data setting may have limited significance.

Second, dPL demonstrates the considerable advantage of binding DL training infrastructure with existing process-based geoscientific models via differentiable computing. This integration is especially helpful where a full simulation history involving unobserved variables is needed to provide interpretable narratives or diagnostics, e.g., climate change impact assessments. On the other hand, our approach can immediately and greatly boost the accuracy of large-scale, socio-economically important geoscientific models such as the National Water Model[46], which is responsible for predicting floods on a national scale. A core enabling technique we proposed here is differentiable computing via either the use of a differentiable surrogate model (as for VIC) or reimplementation on DL platforms (as for HBV). Both options are supported by most modern DL platforms, and a choice can be made depending on the effort level. For simpler models like HBV, it is likely easiest and most accurate to directly implement it on a DL platform, but for more complex or expensive models like VIC, a surrogate model may be the bridge between them and big data machine learning. Surrogate models add one more layer of complexity, however, so a direct implementation is preferred if possible. Following this integration, there should be many pathways towards leveraging machine learning to improve our physical understanding such as learning about better structures in the model. It should be more and more evident that the differentiable computing paradigm sports enormous advantages over the traditional ones when it comes to learning knowledge.

Third, DL-supported dPL offers a generic, adaptive, and highly efficient solution to a large variety of models in geosciences and beyond. For our three examples, each with some different configurations, we used the same $g_A$ and $g_Z$ network components with little customization. We expect such genericity to carry over to other domains. This is because we do not explicitly specify any transfer functions: these are determined by the DL algorithm, which can adapt to new problems automatically. In contrast, for regionalization schemes, new transfer functions need to be conceived for every new model, new calibrated parameters, or even new experiments, as shown in this work with 8-attribute vs 27-attribute model versions. dPL is more flexible than traditional methods to stand up to the challenges of widely different datasets and problem formulations (e.g., traditional regionalization methods cannot use meteorological time series data as raw inputs for parameter inference). With our end-to-end framework, the ability to avoid "ground truth" parameters also leads to performance gains because these parameters are ambiguous (ground-truth parameters do not exist), and adds significant workload.

Fourth, no work in geosciences, to the best of our knowledge, has proposed a structure like $g_Z$, which is an attempt to learn the mapping from historical forcing and response pairs to model parameters. $g_A$: $\mathbf{A} \rightarrow \theta$ and $g_Z$: $(\mathbf{A'}, \mathbf{x}, \mathbf{z}) \rightarrow \theta$ each has their use cases. If we have

good-quality and problem-relevant inputs as $\mathbf{A}$ and $\mathbf{A'}$, then $g_Z$ is not expected to have noticeable advantages over $g_A$ for an in-training site, because the information of $\mathbf{z}$ is already implicitly used during training. Indeed, $g_Z$ showed only a mild advantage for the soil moisture spatial neighbor test. However, in cases where the attributes are limited, i.e., there is not a well-defined mapping from $\mathbf{A'}$ to $\theta$ but we have $\mathbf{z}$ (in an extreme case, we may not have useful $\mathbf{A'}$ at all), then $g_Z$ may be more valuable. An example may be ecosystem modeling, where we have ample observations of top-canopy variables such as leaf area index, but forest species, successional stages, and understory communities lack detailed data except at a small number of sites. We caution, however, that it is still too early to conclude on the general usefulness of $g_Z$ for other problems and more tests are needed to verify the existence, robustness (with respect to data noise), and value of this mapping.

$g_Z$ also possesses some unique advantages in terms of data privacy. There are many places in the world, e.g., China, India, and even privately-owned land in the US like agricultural farms, where, for various reasons, stakeholders do not support data sharing. As $g_Z$ uses local private data as an input, it permits the use of data that do not have the option to participate in training, and can avoid expensive re-training for small incremental datasets. An added benefit is that once trained, the network can be saved and applied at negligible computational cost to new instances, which is not possible for traditional paradigms; inferring parameters for the entire CONUS at high resolution using dPL takes mere seconds.

dPL's advantages are conditional. For cases where one has a limited dataset from only one or a few sites, dPL may not have advantages over traditional approaches, but differentiable computing should still be useful. Another situation where the effectiveness of $g_A$ may not manifest itself is when the inputs do not capture significant variations in the underlying processes (called latent processes). This would require $g_A$ to learn an ambiguous or ill-defined mapping between inadequate inputs and the PBM parameters. In the case of streamflow modeling where geology is a poorly-described latent process, we have not noticed a significant impact. However, one needs to be aware of such potential pitfalls.

dPL can be interpreted as imposing the PBM as a physical constraint for the parameter estimation network component in order to produce parameters that are sensible for the PBM, and that by doing so, physical meaning is attached to multiple outputs of the network. Depending on the setup, imposing physical constraints has been shown to improve generalization[34,47] and certainly builds an important bridge between process knowledge and data science. Overall, the DL-based dPL approach showed immense advantages in efficiency, performance, and robustness over traditional methods.

## Methods

**General description of a geoscientific model and parameter calibration.** A model for both non-dynamical and dynamical systems can be generically written for site $i$ as

$$\{\mathbf{y}_t^i\}_{t \in T} = f\left(\{\mathbf{x}_t^i\}_{t \in T}, \boldsymbol{\varphi}^i, \boldsymbol{\theta}^i\right) \qquad (1)$$

where output physical predictions ($\mathbf{y}_t^i = \left[y_{1,t}^i, y_{2,t}^i, \cdots\right]^T$, with the first subscript denoting variable type) vary with time ($t$) and location ($i$), and are functions of time- and location-specific inputs ($\mathbf{x}_t^i = \left[x_{1,t}^i, x_{2,t}^i, \cdots\right]^T$), location-specific observable attributes ($\boldsymbol{\varphi}^i = \left[\varphi_1^i, \varphi_2^i, \cdots\right]^T$), and location-specific unobserved parameters that need to be separately determined ($\boldsymbol{\theta}^i = \left[\theta_1^i, \theta_2^i, \cdots\right]^T$). $\boldsymbol{\theta}$ may be unobservable, or it may be too expensive or difficult to observe at the needed accuracy, resolution, or coverage. This formulation also applies to dynamical systems if $\mathbf{x}_t^i$ includes previous system states $\mathbf{y}_{t-1}^i$ (i.e. $\mathbf{y}_{t-1}^i \mathbf{x}_t^i$), and the rest of the inputs are independent (e.g. meteorological) forcing data. In a non-dynamical system, $\mathbf{x}_t^i$ is independent of $\mathbf{y}_{t-1}^i$.

Given some observations

$$\mathbf{z}_t^i = h(\mathbf{y}_t^i) + \boldsymbol{\varepsilon}_t^i \tag{2}$$

where $h(\cdot)$ relates model outputs to observations and $\boldsymbol{\varepsilon}_t^i = \left[\varepsilon_{1,t}^i, \varepsilon_{2,t}^i, \cdots\right]^T$ is the error between the observations $\left(\mathbf{z}_t^i = \left[z_{1,t}^i, z_{2,t}^i, \cdots\right]^T\right)$ and the model predictions $(\mathbf{y}_t^i)$, we adjust the model parameters so that the predictions best match the observations. This is traditionally done individually for each location (here generically referring to a gridcell, basin, site, river reach, agricultural plot, etc., depending on the model):

$$\hat{\theta}^i = \arg\min_{\theta^i} \sum_{t\in T} \|\boldsymbol{\varepsilon}_t^i\|^2 = \arg\min_{\theta^i} \sum_{t\in T} \|h(f(\{x_t^i\}_{t\in T}, \varphi^i, \theta^i)) - z_t^i\|^2 \tag{3}$$

where $i \in I$ and where $I = \{1, 2, \ldots, N_I\}$. Note that the superscript $i$ suggests that this optimization is done for each site independently.

**The process-based hydrologic model and its surrogate.** The Variable Infiltration Capacity (VIC) hydrologic model has been widely used for simulating the water and energy exchanges between the land surface and atmosphere, along with related applications in climate, water resources (e.g., flood, drought, hydropower), agriculture, and others. The model simulates evapotranspiration, runoff, soil moisture, and baseflow based on conceptualized bucket formulations. Inputs to the model include daily meteorological forcings, non-meteorological data, and the parameters to be determined. Meteorological forcing data include time series of precipitation, air temperature, wind speed, atmospheric pressure, vapor pressure, and longwave and shortwave radiation. More details about VIC can be found in Liang et al.[39].

LSTM was trained to reproduce the behavior of VIC as closely as possible while also allowing for gradient tracking. In theory, if a hydrologic model can be written into a machine learning platform (as in our HBV case), this step is not needed, but training a surrogate model is more convenient when the model is complex. To ensure the surrogate model had high fidelity in the parameter space where the search algorithms want to explore, we iterated the training procedure multiple times. We first trained an LSTM surrogate for VIC using the forcings, attributes, and parameters from NLDAS-2 as inputs, and the VIC-simulated surface soil moisture (variable name: SOILM_lev1) and evapotranspiration (ET, variable name: EVP) as the targets of emulation. Then, as the search algorithms (SCE-UA or dPL) went near an optimum, we took the calibrated parameter sets, made perturbations of them by adding random noise to these parameters, and retrained the network with added data. The perturbation was done to better represent the parameter space close to optimal solutions. We repeated this procedure four times so that the NSEs of the parameters, obtained from the CPU-based VIC model, converged. At $1/8^2$ sampling density (sampling one gridcell from each $8 \times 8$ patch), this results in fewer overall forward runs than a 1/8-degree NLDAS-2 simulation. Also note that this effort is needed similarly for both dPL and SCE-UA. If we did not use the surrogate model, SCE-UA would also have needed to employ the $O(10^2)$ more expensive CPU-based VIC model. We evaluated the accuracy of the surrogate model, and the median correlations between VIC and the surrogate simulation were 0.91 and 0.92 for soil moisture and ET, respectively (Supplementary Fig. S2). When we connected the trained surrogate model to the parameter estimation network, the weights of the surrogate model were frozen and prevented from updating by backpropagation, but the gradient information could pass through. This was implemented in the PyTorch deep learning framework[36].

**The long short-term memory network.** The long short-term memory network (LSTM) was originally developed in the artificial intelligence field for learning sequential data, but has recently become a popular choice for hydrologic time series data[26]. As compared to a vanilla recurrent neural network with only one state, LSTM has two states (cell state, hidden state) and three gates (input gate, forget gate, and output gate). The cell state enables long-term memory, and the gates are trained to determine which information to carry across time steps and which information to forget. These units were collectively designed to address the notorious DL issue of the vanishing gradient, where the accumulated gradients would decrease exponentially along time steps and eventually be too small to allow effective learning[48]. Given inputs $I$, our LSTM can be written as the following:

$$\text{Input transformation}: \quad x^t = ReLU(W_I I^t + b_I) \tag{4}$$

$$\text{Input node}: \quad g^t = \tanh(\mathscr{D}(W_{gx}x^t) + \mathscr{D}(W_{gh}h^{t-1}) + b_g) \tag{5}$$

$$\text{Input gate}: \quad i^t = \sigma(\mathscr{D}(W_{ix}x^t) + \mathscr{D}(W_{ih}h^{t-1}) + b_i) \tag{6}$$

$$\text{Forget gate}: \quad f^t = \sigma(\mathscr{D}(W_{fx}x^t) + \mathscr{D}(W_{fh}h^{t-1}) + b_f) \tag{7}$$

$$\text{Output gate}: \quad o^t = \sigma(\mathscr{D}(W_{ox}x^t) + \mathscr{D}(W_{oh}h^{t-1}) + b_o) \tag{8}$$

$$\text{Cell state}: \quad s^t = g^t \odot i^t + s^{t-1} \odot f^t \tag{9}$$

$$\text{Hidden state}: \quad h^t = \tanh(s^t) \odot o^t \tag{10}$$

$$\text{Output}: \quad y^t = W_{hy}h^t + b_y \tag{11}$$

where $W$ and $b$ are the network weights and bias parameters, respectively, and $\mathscr{D}$ is the dropout operator, which randomly sets some of the connections to zero. The LSTM network and our whole workflow[31] were implemented in PyTorch[36], an open source machine learning framework.

Here we do not use LSTM to predict the target variable. Rather, LSTM is used to (optionally) map from time series information to the parameters in our $g_z$ network as described below.

**The parameter estimation network.** We present two versions of the dPL framework. The first version allows us to train a parameter estimation network over selected training locations $I_{\text{train}}$ where some ancillary information $\mathbf{A}$ (potentially including but not limited to attributes in $\boldsymbol{\varphi}^i$ used in the model) is available, for training period $T_{\text{train}}$ (illustrated in Fig. 1b):

$$\hat{\theta}^i = g_A(\mathbf{A}^i) \text{for all } i \in I_{\text{train}} \tag{12a}$$

$$\hat{g}_A(\cdot) = \arg\min_{g_A(\cdot)} \sum_{t\in T, i\in I_{\text{train}}} \|h(f(x_t^i, \varphi^i, g_A(\mathbf{A}^i))) - z_t^i\|^2 \tag{12b}$$

Essentially, we train a network ($g_A$) mapping from raw data ($\mathbf{A}$) to parameters ($\theta$) such that the PBM output ($f$) using $\theta$ best matches the observed target ($\mathbf{z}$). We are not training to predict the observations – rather, we train $g_A$ on how to best help the PBM to achieve its goal. The difference between Eq. 12 and Eq. 3 highlights that the loss function combines the sum of squared differences for all sites at once.

The second version is applicable where some observations $\{z_t^i\}_{t\in T}$ are also available as inputs at the test locations:

$$\hat{\theta}^i = g_z\left(\{\mathbf{x}_t^i\}_{t\in T}, \mathbf{A}'^{,i}, \{\mathbf{z}_t^i\}_{t\in T}\right) \text{for all } i \in I_{\text{train}} \tag{13a}$$

$$\hat{g}_z(\cdot) = \arg\min_{g_z(\cdot)} \sum_{t\in T_{\text{train}}, i\in I_{\text{train}}} \|h(f(x_t^i, \varphi^i, g_z(\{\mathbf{x}_t^i\}_{t\in T}, \mathbf{A}'^{,i}, \{z_t^i\}_{t\in T}))) - z_t^i\|^2 \tag{13b}$$

Essentially, we train a network ($g_z$) that maps from attributes ($\mathbf{A}'$), historical forcings ($\mathbf{x}$), and historical observations ($\{z_t^i\}_{t\in T}$) to a suitable parameter set ($\theta$) with which the PBM output best matches the observed target ($\mathbf{z}$) across all sites in the domain. Ancillary attributes $\mathbf{A}'$ may be as or different from $\mathbf{A}$ used in $g_A$, and in the extreme case may be empty. Succinctly, they can be written as two mappings, $g_A: \mathbf{A} \rightarrow \theta$ and $g_z: (\mathbf{A}', \mathbf{x}, \mathbf{z}) \rightarrow \theta$. $g_Z$ can accept time series data as inputs and here we choose LSTM as the network structure for this unit. There is no circular logic or information leak because the historical observations ($\{z_t^i\}_{t\in T}$) are for a different period ($T$) than the main training period ($T_{\text{train}}$). In practice, this distinction may not be so crucial as the PBM acts as an information barrier such that only values suitable as parameters ($\theta$) can produce a reasonable loss. As LSTM can output a time series, the parameters were extracted only at the last time step. For $g_A$, only static attributes were employed, and so the network structure amounts to a multilayer perceptron network. After some investigation of training and test metrics, we set the hidden size of $g$ to be the same as for the surrogate model.

The whole network is trained using gradient descent, which is a first-order optimization scheme. Some second-order schemes like Levenberg–Marquardt often have large computational demand and are thus rarely used in modern DL[49]. To allow gradient accumulation and efficient gradient-based optimization and to further reduce the computational cost, we can either implement the PBM directly into a differentiable form, as described in the global PUB case below, or first train a DL-based surrogate model $f'(\bullet) \simeq f(\bullet)$ and use it in the loss function instead of $f(\cdot)$,

$$g(\cdot) = \arg\min_{g(\cdot)} \sum_{t\in T_{\text{train}}, i\in I_{\text{train}}} \|h(f'(x_t^i, \varphi^i, g(\cdot))) - z_t^i\|^2 \tag{14}$$

where $g(\bullet)$ generically refers to either $g_A$ or $g_Z$ with their corresponding inputs. $g_A$ can be applied wherever we can have the ancillary inputs $\mathbf{A}$, while $g_Z$ can be applied over areas where forcings and observed responses ($\mathbf{x}$, $\mathbf{z}$) are also available, without additional training:

$$\hat{\theta}^i = g_z(\{\mathbf{X}_t^i\}_{t\in T}, \boldsymbol{\varphi}^i, \{\mathbf{Z}_t^i\}_{t\in T}) \text{ or } \hat{\theta}^i = g_A(\boldsymbol{\varphi}^i) \text{ for any } i \text{ and any reasonable } T \tag{15}$$

We tested both $g_A$ and $g_z$, which work with and without forcing-observation ($x$-$z$) pairs among the inputs, respectively. Since SMAP observations have an irregular revisit schedule of 2–3 days and neural networks cannot accept NaN inputs, we have to fill in the gaps, but simple interpolations do not consider the effects of rainfall. Here we used the near-real-time forecast method that we developed earlier[30]. Essentially, this forecast method uses forcings and integrates recently available observations to forecast the observed variable for the future time steps, achieving very high forecast accuracy (ubRMSE < 0.02). When recent observations are missing, their places are taken by the network's own predictions, thus introducing no new information. Using this method, we generated continuous SMAP observations as $z$ for network $g_Z$.

As with most DL work, the hyperparameters of dPL needed to be adjusted. We manually tuned hidden sizes and batch size using one year of data (2015-04-01 to 2016-03-31) using mostly the $1/16^2$ sampling density (sampling one gridcell from each $16 \times 16$ patch). Higher sampling densities led to larger training data, which could be better handled by larger hidden sizes. For sampling densities of $1/16^2$, $1/8^2$, and $1/4^2$ we used hidden sizes of 64, 256, and 1280, respectively. We used a batch size of 300 instances and the length of the training instances was 240 days. Given 1 or 2 years' worth of training data, the code randomly selected gridcells and

time periods with a length of 240 days within the training dataset to form a minibatch for training (In contrast, SCE-UA uses all available years of training data at once, as this is the standard approach). The network's weights were updated after computing the combined loss of each minibatch. We employed the AdaDelta network optimization algorithm[50], for which the coefficient used to scale delta before applying it to the parameters was 0.5, the coefficient used for computing a running average of squared gradients was 0.95, and the weight decay was 0.00001. The dropout rate for the $g_Z$ or $g_A$ component of the network (Fig. 1) was 0.5. We normalized inputs and outputs by their CONUS-wide standard deviation.

Based on Troy et al.[44], the calibrated parameters include the variable infiltration curve parameter (INFILT), maximum base flow velocity (Dsmax), fraction of maximum base flow velocity where nonlinear base flow begins (Ds), fraction of maximum soil moisture content above which nonlinear baseflow occurs (Ws), and variation of saturated hydraulic conductivity with soil moisture (EXPT). INFILT decides the shape of the Variable Infiltration Capacity (VIC) curve and denotes the amount of available infiltration capacity. The formula regarding INFILT in VIC is:

$$i = i_m[1 - (1 - a_f)^{\frac{1}{\text{INFILT}}}] \tag{16}$$

where $i$ is infiltration capacity, $i_m$ is maximum infiltration capacity, and $a_f$ is the fraction of saturated area. With other parameters and $a_f$ being the same, larger INFILT leads to a reduced infiltration rate and thus higher runoff.

For all models, we collected atmospheric forcing data from the North American Land Data Assimilation System phase-II dataset (NLDAS-2)[51]. A number of static physiographic attribute inputs were added to provide additional context for the module to better understand the input-response relationships and correctly estimate the parameters. These attributes included bulk density, soil water holding capacity, and sand, silt, and clay percentages from the International Soil Reference and Information Centre - World Inventory of Soil Emission Potentials (ISRIC-WISE) database[52]. Other inputs were SMAP product flags indicating mountainous terrain, land cover classes, urban areas, and fraction of land surface with water (time averaged).

**SMAP data.** We used the SMAP enhanced level-3 9-km resolution surface soil moisture product (see acknowledgements for access). SMAP level-3 data has an irregular revisit time of 2–3 days, which means there are irregular and densely-distributed gaps in the data. As discussed earlier, we employed the LSTM-based 1-day-ahead forecast scheme to fill the gaps[30]. This scheme utilizes meteorological data and the most-recently-available SMAP observations to provide a near-real-time soil moisture forecast. Essentially, this scheme is a deep-learning version of data assimilation, and has achieved a very low CONUS-scale unbiased RMSE of 0.027, which revised our understanding of the random component of the SMAP data. Using this scheme, we were able to fill the gaps between SMAP observations, and provide seamless data.

To resolve the intrinsic difference between VIC-simulated soil moisture and SMAP, we followed the data assimilation literature[53] and used dPL to estimate, along with other parameters, a linear function with a scaling term ($a$) and a bias term ($b$) between the two:

$$y_{\text{SMAP}} = a * y_{\text{sur}} + b \tag{17}$$

It is widely known that bias correction needs to be applied before observations can be assimilated through data assimilation[54]. $a$ and $b$ are estimated along with the rest of the parameters by dPL and SCE-UA.

**Satellite-based estimates of ET.** MOD16A2[55] is an 8-day composite ET product at 500-meter resolution, which is based on the Penman-Monteith equation. With this algorithm, MODIS 8-day fraction of photosynthetically active radiation is used as the fraction of vegetation cover to allocate surface net radiation between soil and vegetation; MODIS 8-day albedo and daily meteorological reanalysis data are used to calculate surface net radiation and soil heat flux; and MODIS 8-day leaf area index (LAI) and daily meteorological reanalysis data are used to estimate surface stomatal conductance, aerodynamic resistance, wet canopy, soil heat flux, and other environmental variables. MODIS land cover is used to specify the biome type for each pixel to retrieve biome-dependent constant parameters.

We did not use MOD16A2 as a learning target; the purpose here was to validate which calibration strategy led to better descriptions of overall model dynamics. MOD16A2 is not perfect, but since these are completely independent observations from those by SMAP, better agreement should still indicate better modeling of physics overall.

**Shuffled Complex Evolution for comparison.** For comparing the parameter estimation module in dPL, the SCE-UA method[11] introduced three decades ago but still relevant today[56], was also implemented as a reference method. We chose SCE-UA for comparison because it is well established and widely applied. The algorithm ranks a population based on the objective function, and partitions a population of parameter sets into multiple subpopulations called complexes. In one iteration of SCE-UA, the complexes are evolved individually for a number of competitive evolution steps, where reflection, contraction, and random trials are attempted, before they are shuffled and redivided into new complexes for the next iteration.

SCE-UA jobs use the data from the whole training period (1 or 2 years). One SCE-UA iteration contains many VIC forward runs. Because SCE-UA uses discrete iterations involving an uneven number of forward runs, it was not as meaningful to compute the best objective function at the end of a fixed number of runs. Instead, across different gridcells, we collected the lowest objective function RMSE achieved from the beginning to the end of each iteration, and took the average of the ending run as the number of runs to report. We tuned the number of complexes of SCE-UA and set it to seven.

**CAMELS streamflow test.** In earlier work, Mizukami et al.[21] calibrated the VIC model using the multiscale parameter regionalization scheme using data from 531 basins in the Catchment Attributes and Meteorology for Large-Sample Studies (CAMELS) dataset over the contiguous US (see basin locations in Supplementary Fig. S4a). They limited their scope to basins <2000 km² and trained and tested on the same basins, making the experiment a test on the model's ability to generalize temporally. The calibration period of MPR was from 1 October 1999 to 30 September 2008, and the validation period was from 1 October 1989 to 30 September 1999. They calibrated transfer functions for 8 default VIC parameters and added two additional parameters (shape and timescale) for routing, which accounts for the time it takes for water to travel from the catchment to the outlet in a mass-conservative manner:

$$\gamma(t : a, \tau) = \frac{1}{\Gamma(a)\tau^a} t^{a-1} e^{-\frac{t}{\tau}} \tag{18}$$

where $t$ is time [T], $a$ is a shape parameter [–] ($a > 0$), $\Gamma()$ is the gamma function, and $\tau$ is a timescale parameter [T][57].

Convolution of the gamma distribution with the runoff depth series is used to compute the fraction of runoff at the current time, which is discharged to its corresponding river segment at each future time as follows:

$$q(t) = \int_0^{t\max} \gamma(s : a, \tau) * R(t - s)ds \tag{19}$$

where $q(t)$ is delayed runoff or discharge [L³T⁻¹] at time step $t$ [T], $R$ is the model-simulated runoff from the basin [L³T⁻¹], and $t$max is the maximum time length for the gamma distribution [T]. To compare with Mizukami et al.[21], we used the same dataset, basins, and training and test periods. They reported a median NSE of 0.30 for VIC.

**Global PUB test.** Beck20 presented a global-scale hydrologic dataset containing forcings, static attributes, and daily streamflow data from 4229 headwater basins across the world. They used a state-of-the-art regionalization scheme for prediction in ungauged basins (PUB), in which no data from test basins were used in the training dataset, thus testing the scheme's capability to generalize spatially. Eight attributes were used for the transfer functions in Beck20, including humidity, mean annual precipitation, mean annual potential evaporation, mean normalized difference vegetation index, fraction of open water, slope, and percentages of sand and clay. They trained linear parameter transfer functions from raw predictors to 12 free parameters of a simple hydrologic model, HBV. In all, 4229 basins were divided into three climate groups: (i) tropical, (ii) arid and temperate, and (iii) cold and polar. Transfer functions were trained for each of these groups. They ran cross validation within each group, e.g., for the arid and temperate group, they further divided the data into 10 random folds, trained transfer functions in 9 of the 10 folds, and tested the transfer functions on the 10th holdout fold; then they rotated to other folds as the holdout data and reported the average metrics from these holdout basins. Because the holdout basins were randomly selected, there will always be neighboring basins that were included in the training set. However, the sparser the training data are, the further away the holdout basins will be, on average, from the training basins. Hence, we can reduce the number of training basins to examine the impacts of less training data on the model's ability to generalize in space.

Since the primary purpose of this part of the work was to compare dPL to Beck20's regionalization scheme, we kept the setup as similar as possible and focused on the comparison with their temperate catchment group (see their locations in Supplementary Fig. S4b), for which they reported a median Kling Gupta model efficiency (KGE, see below) coefficient of 0.48. We similarly ran 10-fold cross validation by training $g_A$ (here A includes atmospheric forcings) on 9 randomly chosen groups of basins and testing it on the 10th, then rotating the holdout group. The training and testing periods were both 2000–2016, the same as used by Beck20. We compared using 8 raw attributes (to be comparable to Beck20) as well as using all 27 available attributes to demonstrate the extensibility of the dPL scheme. To show how the test results scaled with data, we ran additional experiments where the training basin density was systematically reduced with testing basins unchanged. We could afford to do this with dPL using a highly generic procedure. Unfortunately, it was not practical to do this for Beck20's regionalization scheme for comparison, because it was too computationally expensive and labor intensive. Beck20 calibrated the model on the regular gridcells while dPL is applied at the basin scale. The chosen basins are small in area, hence this difference should have marginal impacts. In fact, comparison to past research [Beck 2016] suggests this discretization gives better metrics than the basin-based approach, meaning that this difference, if anything, is biased against dPL.

**Evaluation metrics**. Four statistical metrics are commonly used to measure the performance of model simulation: bias, correlation (Corr), unbiased root-mean-square error (ubRMSE), and Nash-Sutcliffe model efficiency coefficient (NSE). Bias is the mean difference between modeled and observed results. ubRMSE is the RMSE calculated after the bias (systematic model error) is removed during the calculation, and measures the random component of the error. Corr assesses if a model captures the seasonality of the observation, but did not care about the correlation. NSE also considers bias and is 1 for a perfect model but can be negative for poor models. $\bar{y}$ is the average modeled (predicted) value of all pixels, and $\bar{y^*}$ is the average observed value of all pixels.

$$\text{Bias} = \frac{\sum_{i=1}^n (y_i - y_i^*)}{n} \tag{20}$$

$$\text{ubRMSE} = \sqrt{\frac{\sum_{i=1}^n [(y_i - \bar{y}) - (y_i^* - \bar{y^*})]^2}{n}} \tag{21}$$

$$\text{Corr} = \frac{\sum_{i=1}^n [(y_i - \bar{y})(y_i^* - \bar{y^*})]}{\sqrt{\sum_{i=1}^n [(y_i - \bar{y})^2]}\sqrt{\sum_{i=1}^n [(y_i^* - \bar{y^*})^2]}} \tag{22}$$

$$\text{NSE} = 1 - \frac{\sum_{i=1}^n (y_i^* - y_i)^2}{\sum_{i=1}^n (y_i^* - \bar{y^*})^2} \tag{23}$$

All metrics were reported for the test period. When we evaluated $g_Z$ on the training locations, historical observations during the training period ($\{\mathbf{z}_t^i\}_{t\in T}$) were included in the inputs. Then we used the parameters calibrated from the training period to run the model in the test period to get the reported metrics.

For the spatial generalization tests, we sampled one gridcell out of each 8x8 patch (sampling density of $1/8^2$ or s8), and we ran the trained dPL model on another gridcell in the patch (three rows to the north and three columns to the east of the training gridcell). For SCE-UA, we took the parameter sets from the nearest trained neighbor. For $g_A$, we sent in static attributes from the test neighbor to infer the parameters, which were evaluated over the test period. For $g_Z$, we sent in static attributes as well as forcings and observed soil moisture from the test neighbor, but from the training period. All models were evaluated over the test period.

To be comparable to Beck20, we also calculated the Kling-Gupta model efficiency coefficient (KGE), which is similar in magnitude to NSE:

$$\text{KGE} = 1 - \sqrt{(r-1)^2 + (\beta-1)^2 + (\gamma-1)^2} \tag{24}$$

$$\beta = \frac{\mu_s}{\mu_o} \text{ and } \gamma = \frac{\sigma_s/\mu_s}{\sigma_o/\mu_o} \tag{25}$$

where $r$ is the correlation coefficient between simulations and observations, $\beta$ and $\gamma$ are respectively the bias and variability ratio, $\mu$ and $\sigma$ are the mean and standard deviation of runoff, and indices $s$ and $o$ represent simulations and observations.

## Data availability
SMAP L3 data[58] can be downloaded at https://doi.org/10.5067/T90W6VRLCBHI. NLDAS-2 forcing data[59] can be downloaded at https://doi.org/10.5067/6J5LHHOHZHN4. CAMELS data[60] can be downloaded at https://doi.org/10.5065/D6MW2F4D.

## Code availability
The code for dPL with example datasets is available from the permanent web archival service Zenodo at https://doi.org/10.5281/zenodo.5227738.

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

## Acknowledgements
Primary funding was provided to W.P.T. and C.S. by the Office of Biological and Environmental Research of the U.S. Department of Energy under contract DE-SC0016605. D.F. was supported by National Science Foundation Award EAR #1832294. C.S. was also partially supported by National Science Foundation Award OAC #1940190. Besides the above grants, computing resources were partially provided by National Science Foundation Award PHY #2018280 and performed on the Pennsylvania State University's Institute for Computational and Data Sciences' Roar supercomputer.

## Author contributions
W.P.T. ran the VIC-based experiments and wrote an early draft together with C.S.; D.F. ran the HBV-based experiments, M.P. ran some of the VIC model runs; J.L. provided assistance for data preparation and coding; Y.Y. provided advice on running the VIC model; H.B. provided the global headwater catchment dataset and results from a regionalization scheme for comparison; C.S. conceived the study and revised the manuscript. M.P., K.L., D.F., H.B., and C.S. edited the manuscript.

## Competing interests
K.L. and C.S. have financial interests in HydroSapient, Inc., a company which could potentially benefit from the results of this research. This interest has been reviewed by the University in accordance with its Individual Conflict of Interest policy, for the purpose of maintaining the objectivity and the integrity of research at The Pennsylvania State University.
