## [Peer Review File · Nature Communications]

From calibration to parameter learning: Harnessing the scaling effects of big data in geoscientific modelingREVIEWER COMMENTS

Reviewer #1 (Remarks to the Author):

This paper presents a fundamentally new approach for parameter estimation in large-domain hydrological models. The approach is general and applicable across the geosciences. In my opinion, the paper represents the most important advance in large-domain parameter estimation since Luis Samaniego published his MPR method in 2010.

To be more specific, this paper solves one of the most challenging problems in the geosciences: parameter non-uniqueness (also called equifinality by some). Parameter non-uniqueness means that different parameter sets can result in similar outputs, and it can be nearly impossible to obtain meaningful parameter values by calibration. The approach presented here -- spatial learning -- where similar model elements learn from each other, provides a natural physical constraint that substantially improves the inference of parameter values. The approach cleverly uses the nature of big data to constrain the parameter inversion.

I personally found the paper to be thought-provoking. I wanted to stop what I was doing and try a few things out. I expect that this paper will influence thought in large-domain parameter estimation for many years to come. I hope that this paper is published quickly so that others in the community can benefit from these advances in modelling capabilities.

I have some minor comments to improve the presentation:

- Line 21. Grammar. The text should read "The behavior and skill..."
- Line 22. Logic. An uncalibrated model or a poorly calibrated model can also propagate information from observations to unobserved variables via model physics. Please revise.
- Line 26 (and elsewhere). I really do not like the term virtuous. For me, this defines high moral standards -- an attribute that we assign to people, not models. Please revise.
- Line 38 (and elsewhere) you use the term land surface hydrologic models. I find this confusing because many processes are below the land surface. I think it is more precise to say "terrestrial systems (land) models that are used in Earth System Models, hydrological models, ..."
- line 46: Behavior and skill again (needs to be singular)
- Line 46. I do not understand what you mean by "yet underdetermined parameters" (especially the choice of the word "yet"). Should it be "and underdetermined parameters"?
- Line 61. Please remove the parenthetic statement on a key step in Earth System models -- rainfall-runoff models are not really used by the Earth System modelling community.
- Line 72: Exploit such commonalities.
- Line 75: Widely - wildly?
- Line 76: Parameter sets give  parameter sets produce?
- Line 78: Land surface hydrologic models again.
- Line 80: Large scale  large domain. Scale defines the spatial resolution not the spatial extent.
- Line 84: The structure of the transfer function is...
- Line 98: For the prediction of  predicting
- Line 100: we are still reliant  we still rely
- Line 112: Computationally
- Line 115: provide reduced  reduce
- Line 131-140: Suggest use numbers or a numbered list
- Line 147: Remove comma
- Line 148: forlocations  for locations
- Figure 1. The left hand side of the figure is unclear/messy. Think about ways of conveying the information more clearly.
- Line 200: In a DL platform.
- Figure 4. There are other parameter regionalization methods that will provide spatially smooth parameters (e.g., global calibration using multipliers, MPR, etc.). Please clarify that these results can

also be produced using other methods.

- Line 312. Please check the MPR results from Rakovec, Mizukami et al. in JGR atmospheres (<https://doi.org/10.1029/2019JD030767>)
- Line 335: Is material the right word?
- Line 356: Virtuous again.
- Line 416: Virtuous again.
- Line 419: Can you use the word significant in the absence of significance tests?
- Line 451-463: Move this explanation to the description of Figure 1.
- Line 489: Such potential pitfalls.
- Line 578: New paragraph.

Reviewer #2 (Remarks to the Author):

Dear Authors:

I commend you for a very nice piece of work. I found the paper to be well-structured and written, and very easy to read and comprehend. The authors have done a good job of placing the problem and their strategy for addressing it in context – to which I have tried to add my own perspective above. Overall, I thoroughly enjoyed reading the paper, and my comments below by separate attachment are in the spirit of attempting to help them maximize the value of this manuscript to the Geoscientific community.

Overall, I have a very positive assessment of this paper, and definitely recommend publication in Nature Geoscience. By addressing the comments raised above, I believe that science, and the community of geoscientists, would be well served.

I look forward to seeing this manuscript in print. Please feel free to contact me if I can provide any further clarification regarding these review comments.

Please see attached file for my specific review comments

Regards
Hoshin

Reviewer #3 (Remarks to the Author):

The paper presents a framework for learning internal parameters for a process based model through gradient descent with recurrent neural networks. The model scales in performance with more data and produces more spatially consistent parameter estimates. There is also a computational advantage especially when using a surrogate model for training. The broad methodology outlined here has broad application potential beyond hydrology, making Nature Communications a good venue for the paper. My main concern is in the lack of clarity in the presentation, especially the figures. I recommend major revisions to address these issues.

Major Comments

Figure 1: this figure is hard to interpret without reading both the paper and the long caption. Given that this is the key summary diagram for the paper, it should clearly and concisely show the model components and how data and information flow through them. The main issues in the current figure: 1. The sub parts should be annotated with short titles to clearly delineate the focus of each part without having to refer to the caption.

2. The legend is mostly redundant since the boxes are already labeled in the diagrams. The meaning of the additional words can be inferred from the positions of the boxes.
3. Instead of using the term forcing, say dynamic inputs and provide a couple of concrete examples as either words or pictograms.
4. For static attributes also provide concrete examples.
5. Make e and f a separate figure so you have more room for describing the diagrams.
6. Is generic parameter meant to be an output of ga and gz? It should be on the line rather than off to the side.
7. Annotate the output of the PBM

Lines 616-618: this sentence states that the models are manually tuned regarding test set error and the difference between training and test set error. If test set error is used for model tuning, then the test set is no longer independent of the training process, and a separate test set is needed to provide a robust error estimate. Also, it would be helpful to know how many manual iterations were required to converge on the chosen hyperparameters and how long the iterations took. Was a separate manual tuning process employed for each training data density or was tuning done for only one training set size?

Figure 4: How does the spatial smoothness of the parameter estimates vary across training epochs? Does the model produce this spatially smooth pattern as soon as RMSE reaches a reasonable level or does the network need to fine tune for awhile first?

What are the specifics of the GPU and CPUs used to benchmark the models? How much RAM and vRAM are available?

Minor Comments

Line 90: "a type of neural network with large depth" is an overly vague definition of deep learning. Please clarify that deep learning includes multiple layers that are often specialized to find patterns in the spatial and/or temporal structure of data.

Line 180: it would be helpful to mention the temporal length of the simulation up front and not just in the methods, where I believe 240 days was mentioned. Since this journal has a broad audience, the reader's experience with a single run of a simulation could vary greatly depending on their field of expertise.

Lines 191-192: what kind of CPU and GPU were used for the benchmarking? Please specify in the methods.

Figure 2: Can you place the matching marker next to each time series plot? Can you also use lines with greater contrast than blue and black? It is hard to see the differences between the two models.

Figure 4: does infiltr have units? Also, it would be more readable to use infiltration in the title rather than INFILT. Why is the parameter range for SCE so much larger than for dPL?

Lines 320-331: why use NSE for one experiment and KGE for the other? It would be clearer to pick 1 and stick with it.

Figure 5: I am not sure how to interpret this plot? What should the CDF look like ideally? Is the green line a baseline or what you are trying to aim for?

Figure 6: it looks like another significant figure is needed for each label on the green axis.

Line 617: how many models were trained as part of the tuning process? How much computational cost

did this add?

Line 620: do all the networks only use 1 hidden layer with LSTM?

Line 624: how many batches per epoch?

Line 766: should specify that y is the prediction.

Dear editor,

Thank you for handling our manuscript. All three reviewers seemed to favorably view the manuscript. They provided some excellent comments which we leveraged to improve our paper. They did not request major new computational results, but raised some important questions such as a better explanation between g_A vs g_Z , more fair comparison of computational time, train-test split, etc. They asked for many clarifications.

The major changes include

(i) we re-generated Figure 2 (old Figure 1 right column) and replaced the concept of epoch with “number of forward runs”, which is a more fair way of evaluation. Our previous definition of epoch was somewhat confusing. We also added computational time as a comparison. This resulted in changes in appearances in Figure 2, but it did not change our conclusions.

(ii) we ran spatial autocorrelation and test year analysis as requested by reviewer 3.

(iii) we significantly revised the writing of the manuscript.

We believe the revised manuscript should be close to acceptable for publication. Thank you for your consideration.

In the following, numbers in curly brackets {} represent line numbers in the revised clean manuscript.

REVIEWER COMMENTS

Reviewer #1 (Remarks to the Author):

This paper presents a fundamentally new approach for parameter estimation in large-domain hydrological models. The approach is general and applicable across the geosciences. In my opinion, the paper represents the most important advance in large-domain parameter estimation since Luis Samaniego published his MPR method in 2010.

To be more specific, this paper solves one of the most challenging problems in the geosciences: parameter non-uniqueness (also called equifinality by some). Parameter non-uniqueness means that different parameter sets can result in similar outputs, and it can be nearly impossible to obtain meaningful parameter values by calibration. The approach presented here -- spatial learning -- where similar model elements learn from each other, provides a natural physical constraint that substantially improves the inference of parameter values. The approach cleverly uses the nature of big data to constrain the parameter inversion.

I personally found the paper to be thought-provoking. I wanted to stop what I was doing and try a few things out. I expect that this paper will influence thought in large-domain parameter estimation for many years to come. I hope that this paper is published quickly so that others in the community can benefit from these advances in modelling capabilities.

Thank you for your careful consideration of the implications and putting our work in the context of previous work. Below we respond to your comments one by one.

I have some minor comments to improve the presentation:

- Line 21. Grammar. The text should read "The behavior and skill..."

We intentionally used the plural forms of behavior and skill to emphasize that there's not one behavior or skill universally shared by models; each has its own behavior(s) and skill(s).

- Line 22. Logic. An uncalibrated model or a poorly calibrated model can also propagate information from observations to unobserved variables via model physics. Please revise.

*Changed to "can **better** propagate".*

Yes, in a sense uncalibrated or wrongly calibrated models can also propagate information, but less accurately.

- Line 26 (and elsewhere). I really do not like the term virtuous. For me, this defines high moral standards -- an attribute that we assign to people, not models. Please revise.

We initially chose "virtuous" as it is in "virtuous circle" vs the "vicious circle". We think the word succinctly captures the fact that the effect of data scaling is "good" and "increasingly good with more data", and we kinda like the word. However, we understand to some readers this may carry a different meaning.

We changed it to "beneficial" or "advantageous" at different places.

- Line 38 (and elsewhere) you use the term land surface hydrologic models. I find this confusing because many processes are below the land surface. I think it is more precise to say "terrestrial systems (land) models that are used in Earth System Models, hydrological models, ..."

Changed it as suggested to "... such as land models that are used in Earth System Models, hydrologic models that simulate soil moisture, evapotranspiration, runoff, and groundwater recharge" {Line 19-20}

- line 46: Behavior and skill again (needs to be singular)

Please see our response above to the question about line 21

- Line 46. I do not understand what you mean by "yet underdetermined parameters" (especially the choice of the word "yet"). Should it be "and underdetermined parameters"?

Changed to "and" as suggested

- Line 61. Please remove the parenthetic statement on a key step in Earth System models -- rainfall-runoff models are not really used by the Earth System modelling community.

Removed. However, earth system component like Community land model does have the rainfall-runoff routines? In fact, there is a version of CLM that uses VIC as the hydrologic component.

- Line 72: Exploit such commonalities.

Changed as suggested (added "such")

- Line 75: Widely - wildly?

Changed as suggested

- Line 76: Parameter sets give  parameter sets produce?

Changed as suggested

- Line 78: Land surface hydrologic models again.

Changed to "hydrologic model".

- Line 80: Large scale  large domain. Scale defines the spatial resolution not the spatial extent.

Good suggestion. Changed.

- Line 84: The structure of the transfer function is...

We intentionally referred to "transfer functions" to emphasize that there are many possible transfer functions, not one explicit one. For each parameter there needs to be a different function. In fact, in this paper, the plural form was used.

<https://agupubs.onlinelibrary.wiley.com/doi/10.1002/2017RG000581>

- Line 98: For the prediction of  predicting

Changed as suggested

- Line 100: we are still reliant  we still rely

Changed as suggested

- Line 112: Computationally

Changed as suggested

- Line 115: provide reduced  reduce

Changed as suggested

- Line 131-140: Suggest use numbers or a numbered list

Changed as suggested

- Line 147: Remove comma

Changed as suggested

- Line 148: forlocations  for locations

Changed as suggested

- Figure 1. The left hand side of the figure is unclear/messy. Think about ways of conveying the information more clearly.

We have converted this figure into Figure 1 and Figure 2. We paid attention to properly labeling the different frameworks. With the revised version. It should be more clear now.

- Line 200: In a DL platform.

Changed as suggested

- Figure 4. There are other parameter regionalization methods that will provide spatially smooth parameters (e.g., global calibration using multipliers, MPR, etc.). Please clarify that these results can also be produced using other methods.

*Good point. We added a sentence here “While traditional regionalization schemes like MPR are an important and constructive avenue, **e.g., they can also generate spatially-smooth parameter fields looking like Figure 5a**, our comparisons suggest that they are far from optimal and thus cannot fully leverage information provided by big data.” This seems to position this discussion at the right place. {Line 348-351}*

- Line 312. Please check the MPR results from Rakovec, Mizukami et al. in JGR atmospheres

(<https://doi.org/10.1029/2019JD030767>)

Thanks for pointing this paper out. We have read this paper when writing our manuscript. It seems the VIC-related results were carried straight from Mizukami et al., 2017 and that’s why we only cited the Mizukami et al. 2017 paper. We quote it here: “The calibration soil parameters are also identical to Mizukami et al. (2017)” “The results corresponding to the VIC model are identical to those presented by Mizukami et al. (2017).” “Mizukami et al. (2017) discuss the transfer functions implemented in MPR for each VIC soil parameter. In total, 10 transfer function (global,) parameters related to soil parameters are calibrated.” We also confirmed from the figure that the results are identical. Looking at their Figure 2a (pasted below from Rakovec et al., 2019), the VIC CONUS-wide (that is the version calibrated with MPR) has a median daily NSE of 0.32, as we reported here. Hence, it appears our quoted number was accurate.

One thing the reviewer might have meant was that the local model had a higher median NSE of around 0.58. However, as we mentioned earlier, these locally calibrated models tend to be over-calibrated and their internal dynamics and thus the unobserved variables (a main motivation for dPL) might be distorted. We mainly intend to compare with the results from MPR (the CONUS-wide VIC model below).

- Line 335: Is material the right word?

Changed to "consequential"

- Line 356: Virtuous again.

Replied above

- Line 416: Virtuous again.

Replied above

- Line 419: Can you use the word significant in the absence of significance tests?

Changed to "considerable"

- Line 451-463: Move this explanation to the description of Figure 1.

We think this paragraph is "expanded discussion" rather than "explanation" and it fits here better.

- Line 489: Such potential pitfalls.

Changed as suggested

- Line 578: New paragraph.

Changed as suggested

Reviewer #2 (Remarks to the Author):

Dear Authors:

I commend you for a very nice piece of work. I found the paper to be well-structured and written, and very easy to read and comprehend. The authors have done a good job of placing the problem and their strategy for addressing it in context – to which I have tried to add my own perspective above. Overall, I thoroughly enjoyed reading the paper, and my comments below by separate attachment are in the spirit of attempting to help them maximize the value of this manuscript to the Geoscientific community.

Overall, I have a very positive assessment of this paper, and definitely recommend publication in Nature Geoscience. By addressing the comments raised above, I believe that science, and the community of geoscientists, would be well served.

I look forward to seeing this manuscript in print. Please feel free to contact me if I can provide any further clarification regarding these review comments.

Please see attached file for my specific review comments

Regards
Hoshin

[comments from attached file]

Tsai et al, From Calibration to Parameter Learning: Harnessing the Scaling Effects Of Big Data
In Geoscientific Modeling
Manuscript submitted to Nature Geosciences
Review Provided by Hoshin Gupta, The University of Arizona (May 2021)

Executive Summary of the Paper

[1] Problem Statement: This paper addresses the important problem of developing spatially varying parameter fields for large-scale (e.g., regional, continental, global) implementations of spatially-resolved geoscientific models that are based on (and incorporate) representations of domain-specific geoscientific knowledge – e.g., dynamical latent-variable state-space land-surface models that are driven by hydrometeorological forcings.

[2] Limitations of the Traditional Approach: The traditional approach to specifying/infering parameter fields for such models from available information about geometrical, geological and other local properties of the system and by calibration to input-output data is fraught with difficulties and inefficiencies. The reasons for this are numerous and have been well-explored in the literature. They include the fact that the relationships between local system properties and the “parameters” that characterize the parameterization processes (flux equations) are nebulous at the relevant modeling scales and resolutions, due to abstractions, conceptualizations and approximations introduced during the model building process. Further, when calibrating such models to input-output data, the success of the training process can be strongly affected by factors such as data informativeness and completeness,

poorly understood characteristics of the model structural inadequacies (leading to poorly understood biases in model simulations), poor ability of the selected training metrics to extract relevant/signature information from the available data (failure to suitably characterize and distinguish between relevant information and noise), and inefficiencies of the search algorithms used for parameter learning (model training), among others.

[3] One important aspect of the latter issue is that many geoscientific models are coded using equations and algorithmic habits that lead to discontinuities in the derivatives of model fluxes and state variables with respect to the parameters (exemplified algorithmically as “If-Then-Else” statements), and the fact that most such model codes do not include explicit differentiation, so that efficient gradient-based learning algorithms are not easily applicable.

[4] Further, it has historically been common to attempt to “calibrate” such models via a strategy that pursues local (location by location) “parameter value” tuning, rather than by focusing on a strategy that pursues global (over the entire spatial extent) tuning of the spatially coherent “parameter fields”. This strategy can lead to poorly regularized results where the “noise” in the estimates can be (and typically is) so large that it masks the true nature of the spatial patterns in the parameter fields. Recently, various authors (including this reviewer) have attempted to address this problem by imposing “global” regularization constraints on the relationships between local system properties and the parameter fields. This global approach, while demonstrating promise, has been only partially successful due to the difficulty in posing suitable (domain-science-informed) hypotheses regarding the appropriate nature of the regularization constraints, when little understanding of such relationships currently exists. One important reason for the latter is that each version of a geoscientific model hypothesis for a given domain may implement different conceptual representations of the process relationship-and-latent-variable space, so that the so-called “parameters” of such models do not actually “mean” the same thing from one model hypothesis to another. This significantly clouds the waters (so to speak) of geoscientific understanding.

[5] Contribution of The Paper: This paper proposes a comprehensive strategy for addressing the model parameter-specification problem outlined above, by exploiting the power of (1) Big Data, (2) Modern Computational Resources and (3) Machine Learning, specifically modern Deep Learning. Further progress down this path has the potential to realize a significant step-change advance in our ability to develop, train and utilize large-scale spatio-temporal geoscientific models to advance scientific understanding, and to support decision-making.

[6] The main novelty of this new manuscript is to bridge the gaps between Physics-Based Modeling and Machine Learning in a way that exploits the power of both. Their approach to doing this is to use big-data and machine learning to “learn” (rather than hypothesize/assume) the forms of the relationships between local system properties and the “parameters” that characterize the parameterization processes of any given geoscientific model. To enable this, an important step is to enable what the authors term “differentiable parameter learning” (dPL) so that efficient stochastic machine learning (optimization) tools and strategies can be employed. Another important step is to enable the learning process to

choose to exploit, if useful, implicit spatial-parameter-relevant information that may be provided by the input and output fluxes (and their spatiotemporal variations).

[7] Major Findings of The Paper: The results obtained via several experimental studies performed by the authors are very interesting. Importantly, the new approach provides spatially coherent and realistic looking parameter fields that are noticeably less plagued by the “noise” problems inherent to the traditional pointwise calibration approach. Further, the training performance exhibits scaling properties indicating more effective and efficient learning of those spatially-coherent parameter fields, achieving better model performance with orders-of-magnitude lower computational cost, and potentially requiring less training data to achieve performance that equals or exceeds that of the traditional methods.

We thank Dr. Gupta for the excellent discussion of the background and summary of the major contributions. We have attempted to incorporate some of this discussion into our revision.

My Review Comments

[8] I commend the authors for a very nice piece of work. I found the paper to be well-structured and written, and very easy to read and comprehend. The authors have done a good job of placing the problem and their strategy for addressing it in context – to which I have tried to add my own perspective above. Overall, I thoroughly enjoyed reading the paper, and my comments below are in the spirit of attempting to help them maximize the value of this manuscript to the geoscientific community.

[9] The main aspect of the paper that I think can be better explored is the relative value of using the g_A and g_z versions of the dPL framework. Since the g_A version explicitly uses only information provided observable attributes A , (info about forcing response pairs $x-z$ is incorporated implicitly via training of the physical model), while the g_z version explicitly and directly allows the network to access and use information provided by forcing-response pairs $x-z$, it would be interesting to see at least one example that explores the differences and relative benefits of the two somewhat complementary but not entirely distinct approaches. More specifically, since forcing-response pairs $x-z$ are being used anyway to train the model, it is important to examine what additional benefit may be provided by using g_z instead of g_A .

This is a good point. We have several responses. For easiness of response, the two frameworks can also be written succinctly as $g_A: A \rightarrow \theta$ and $g_z: (A', x, z) \rightarrow \theta$. In our original write up, we did not mention that A and A' can be different (we added that explanation now).

1. In some cases you have lots of good-quality and problem-relevant geophysical attributes for A and A' . Then, indeed, we expect g_A and g_z to have little difference for in-training sites. Exactly as Dr. Gupta said, the information in z has been implicitly used.
2. However, consider the situation where the A' we have at hand does not describe the problem well. Think about the most extreme case where we have nothing for A' , i.e., A' is empty and g_z reduces to $(x, z) \rightarrow \theta$. This reverts back to the typical hydrologic model calibration problem

where all information comes from z . In this case, g_z is the only applicable framework and normally such inversion would be plagued by equifinality. Even in this case, applying a global constraint could have beneficial effects.

3. Follow up with point 1 above, if z for a site is already known, why don't we include it in the training dataset? The reality is there are plenty of scenarios where data privacy matters: the model is trained by a group of people while the user may be another group of folks; The training of the model may be too expensive to make it worthwhile for small incremental data points; or due to privacy reasons some users may not be able to contribute their data to the training dataset; it is also possible that the trainers of the model could share the trained model with the world but cannot share the data due to data ownership reasons. The data privacy is not a minor issue in hydrology, as anyone who has hunted for data in China, India or Africa can tell how elusive such data can be. In fact, in the realm of AI (especially in medical AI applications), there is a line of research called federated learning (FL) that studies problems in a privacy-constrained environments (<https://ai.googleblog.com/2017/04/federated-learning-collaborative.html>, https://en.wikipedia.org/wiki/Federated_learning). It would be too much for us to step into FL in this paper (already discussing lots of new ideas) so we did not introduce it. In addition, we believe future work is needed to further study the g_z structure and its robustness. Hence, we have revised two relevant paragraphs to address the reviewer's question.

“Fourth, no work in geosciences, to the best of our knowledge, has proposed a structure like g_z which is an attempt to learn the mapping from historical forcing and response pairs to model parameters. $g_A \rightarrow \theta$ and $g_z: (A', x, z) \rightarrow \theta$ each has their use cases. If we have good-quality and problem-relevant inputs as A and A' , then g_z is not expected to have noticeable advantages over g_A for an in-training site, because the information of z is implicitly used during training. Indeed, g_z showed a mild advantage only for the soil moisture spatial neighbor test. However, in cases where A' is limited (in extreme cases we may not have useful A' at all), i.e., there is not a well-defined mapping from A' to θ but we have z , then g_z should be more valuable. The results of g_z also emphasize the global constraint more than uncaptured local heterogeneity. An example may be ecosystem modeling, where we have ample observations of top-canopy variables such as leaf area index, but forest species, successional stages, and understory communities lack detailed data except at a small number of sites. More tests are needed to verify the existence, robustness (with respect to data noise) and value of this mapping for different problems.

Moreover, g_z possesses some unique advantages in terms of data privacy. There are many places in the world, e.g., China, India, and even privately-owned land in the US like agricultural farms, where, for various reasons, stakeholders do not support data sharing. g_z allows the use of local private data that cannot participate in training, and can avoid expensive re-training for small incremental datasets. An added benefit is that once trained, the network can be saved and applied at negligible computational cost to new instances, which is not possible for traditional paradigms; inferring parameters for the entire CONUS at high resolution using dPL takes mere seconds.” {Line 464-484}

[10] One other point that I think should be at least briefly discussed in the paper is that the performance statistics being reported for most of the cases (e.g., NSE) are relatively “low” ... one might typically aspire to obtain NSE or KGE values above 0.8 and preferably above 0.9 (a subjective choice for sure) to have some kind of confidence in the predictive ability of a model. Because of the strong correspondence between these performance measures and the linear correlation coefficient, lower values tend to indicate large amounts of noise and/or bias, in the simulated-to-observed relationships. One inevitably wonders to what extent the “bias error”, “variability error” and “correlation/timing-shape error” components of model inadequacy are dominating the contributions to poor model performance. This comment is not in any way intended as a critique of your work, but instead to propose that it is important to inform the reader of Nature Geoscience of two things – one being how to make sense of these metric values (how to contextualize them), and the second being to point out that the poor values are not a consequence of the ML paradigm per se, but instead largely arise due to relatively poor understanding of the physics and/or information needed to properly characterize the associated geoscientific system.

Very good point about the performance of the model. One could think of forcing/attribute errors and model structural errors. While LSTM. We originally had some discussion about this, but now will make it more explicit.

Originally “Meanwhile, a pure LSTM model can certainly achieve a higher NSE than VIC³², suggesting there is substantial room to improve VIC structure.” {Line 315-316}

Revised to → “Despite the much improved parameter learning scheme, the best results achievable with VIC (median NSE=0.44) is much weaker than a pure LSTM model, which can obtain a median NSE of 0.74 (with the use of ensemble)³², which may be interpreted as being close to the best-possible model given forcing and attribute errors. The gap from 0.44 to 0.74 can be partially explained by the imperfectness of the surrogate model, but it is unlikely the major culprit based on its agreement with VIC (Figure S1). Given dPL’s strong optimizing capability (previously demonstrated in comparison with SCE-UA), the remaining gap can be mostly attributed to the limited understanding of hydrology encoded in the structures of VIC or HBV, which have large room for improvement. There is a potential in the future to extend our framework to further learn better model structure and, by doing so, improve our understanding of the physical system.” {Line 317-326}

We also revised a sentence to be the following:

“For all cases, the dPL metrics are still considerably lower than what would be obtained by purely data-driven LSTMs, but again, this is likely due the limitations of the PBM structures and the ways the reference problems were set up in the literature, e.g., choices of inputs, no routing in the global PUB case, etc., as. We inherited these setups because the main purpose here is to compare with previous schemes.” {Line 343-348}

[11] Overall, I have a very positive assessment of this paper, and definitely recommend publication in Nature Geoscience. By addressing the comments raised above, I believe that science, and the community of geoscientists, would be well served.

[12] Congratulations again, on a very noteworthy contribution, and I look forward to seeing this manuscript in print. Please feel free to contact me if I can provide any further clarification regarding these review comments.

My Recommendation

[13] Publish with moderate revisions.

Regards

Hoshin

Reviewer #3 (Remarks to the Author):

The paper presents a framework for learning internal parameters for a process based model through gradient descent with recurrent neural networks. The model scales in performance with more data and produces more spatially consistent parameter estimates. There is also a computational advantage especially when using a surrogate model for training. The broad methodology outlined here has broad application potential beyond hydrology, making Nature Communications a good venue for the paper. My main concern is in the lack of clarity in the presentation, especially the figures. I recommend major revisions to address these issues.

Thank you for your fair and constructive evaluation.

Major Comments

Figure 1: this figure is hard to interpret without reading both the paper and the long caption. Given that this is the key summary diagram for the paper, it should clearly and concisely show the model components and how data and information flow through them. The main issues in the current figure:

1. The sub parts should be annotated with short titles to clearly delineate the focus of each part without having to refer to the caption.
2. The legend is mostly redundant since the boxes are already labeled in the diagrams. The meaning of the additional words can be inferred from the positions of the boxes.
3. Instead of using the term forcing, say dynamic inputs and provide a couple of concrete examples as either words or pictograms.
4. For static attributes also provide concrete examples.
5. Make e and f a separate figure so you have more room for describing the diagrams.
6. Is generic parameter meant to be an output of ga and gz? It should be on the line rather than off to the side.
7. Annotate the output of the PBM

Thank you for your detailed suggestions. They are very useful. We have revised the figure as below. We separated out e and f to a separate figure.

Above: New Figure 1.

Above: New Figure 2.

Lines 616-618: this sentence states that the models are manually tuned regarding test set error and the difference between training and test set error. If test set error is used for model tuning, then the test set is no longer independent of the training process, and a separate test set is needed to provide a robust error estimate. Also, it would be helpful to know how many manual iterations were required to converge on the chosen hyperparameters and how long the iterations took. Was a separate manual tuning process employed for each training data density or was tuning done for only one training set size?

The question about train and test is a good one. Yes, with a machine learning workflow, we normally separate the data into train, validation (used to tune hyperparameters) and test (not used at all in anything other than evaluating metrics and visualization). We have several responses:

(1) The spatial test (both soil moisture spatial extrapolation in case 1 and streamflow PUB in case 3) are completely test data that were not involved in training or hyperparameter tuning.

(2) For the soil moisture case, we added a Figure in the Supplementary Information (pasted below) which shows the Figure for a different year (2018-2019) which is completely uninvolved in the training or tuning. One can see that, while the numbers could fluctuate, the relative advantages over SCE-UA remain very similar, i.e., SCE-UA deteriorated from training to test sites while dPL had no deterioration for the test sites (uninvolved in training or tuning).

The main text referred to this Figure “A similar pattern is observed for another year as shown in Figure S5 in Supplementary Information.” {Line 212-213}

(3) Most of the hyperparameters except for the hidden size were tuned based on s16 (from 2015-04-01 to 2016-03-31). This was a very lazy tuning. The hidden sizes of LSTM are tuned for each training data density (but we still used one year training period, from 2015-04-01 to 2016-03-31). The text has been updated to “We manually tuned hidden sizes, batch size using one year of data (2015-04-01 to 2016-03-31) using mostly the s16 sampling density except for hidden sizes” {Line 629-631}

Above: Figure S5, for independent test sets. This figure shows that both dPL g_A and dPL g_z have better performance than SCE-UA in the T2 (the testing period is from 2018.04.01 to 2019.03.31).

Figure 4: How does the spatial smoothness of the parameter estimates vary across training epochs? Does the model produce this spatially smooth pattern as soon as RMSE reaches a reasonable level or does the network need to fine tune for awhile first?

Good question, we calculated spatial autocorrelation for INFILT across different epochs, as shown below. The results indicate that the spatial autocorrelation is kept higher than 0.7 from the beginning to the end. We added this Figure to the Supplementary Information and added a sentence to mention it. *“The spatial autocorrelation was high from the beginning to the end of the optimization (Figure S6a), showing the global constraint allows the model to learn across sites since the beginning.”* {Line 285-287}

Moran's I is a measure of spatial autocorrelation developed by Patrick Alfred Pierce Moran.

Moran's I measures the global spatial autocorrelation in an attribute y measured over n spatial units and is given as:

$$I = \frac{\sum_i \sum_j w_{ij} (y_i - \bar{y})(y_j - \bar{y})}{\sum_i \sum_j w_{ij} (y_i - \bar{y})^2}$$

where w_{ij} is a spatial weight, $w_{ij} = 1$ if i, j are adjacent and 0 otherwise, and $\bar{y} = \frac{1}{n} \sum_i y_i$.

In the Moran Scatterplot (shown as below), Quadrant I means the high-value area is surrounded by high value; Quadrant II means the low-value area is surrounded by high value; Quadrant III means the low-value area is surrounded by low value; Quadrant IV means the high-value area is surrounded by low value. The higher Moran's I is, the higher spatial autocorrelation is.

In contrast to the dPL, the spatial autocorrelation of SCEUA is only 0.11.

What are the specifics of the GPU and CPUs used to benchmark the models? How much RAM and vRAM are available?

The reviewer raised a very good point about getting more precise with the comparison. We agree the previous sentence about a 100-CPU cluster for 2-3 days was too anecdotal. Upon checking, the previous use of “epoch” was somewhat unique and could be confusing and we apologize for that. Because our code randomly samples from available time series (to avoid potential overfitting to fixed starting points), an epoch was defined as corresponding to the probability that 99% of the time periods of all basins are picked in the epoch. However, this complicates the comparison with respect to computation.

To avoid any confusion, we now provide two new comparisons: (1) number of forward runs per gridcell, which is purely a comparison of the number of times the training data have been used; and (2) single CPU vs single GPU computing time. The Figures have been updated to reflect this. The advantage of dPL

in terms of forward runs now seems smaller than the previous figure, but the pattern is still quite clear that higher sampling density results in much fewer runs, and at s4 the forward runs for dPL is much fewer than SCE-UA. The contrast is drastic in computational time.

“Notably, as training data increases, dPL descended into the range of acceptable performance orders of magnitude faster than SCE-UA in terms of either the number of forward runs (Figure 2a-b) or computational time (Figure 2c-d). For the model trained for 2 years, dPL required 930, 250, and 45 full forward runs per gridcell (or 2.5, 0.49, and 0.31 seconds of computing time per gridcell, not proportional to forward runs due to increasing hidden sizes) to drop below the threshold for a functional model (RMSE=0.05) at 1/162, 1/82, and 1/42 sampling densities, respectively. With the same surrogate model running on a GPU, similar for all three sampling densities, SCE-UA’s needs 950 runs per gridcell (or 90 seconds, here we did not implement parallelism for SCE-UA) to reach the same RMSE. Two factors are at play for dPL’s efficiency: the first factor is the reduction of runs at higher sampling densities (with an order of magnitude spread between 1/162 and 1/42 in terms of either runs or time). This super scaling effect resulted from the use of a domain-wide loss function and mini-batch training, allowing dPL to learn across locations rapidly (more interpretation in Discussion). The second factor is the inherent parallelism offered by dPL via modern DL infrastructure. Substantial effort would be needed to enable parallelism by SCE-UA on the GPU and there are some serial simulations that cannot be parallelized. While SCE-UA can also be parallelized, it may be difficult to achieve the high level of parallel efficiency and scale enjoyed by dPL.

While using a surrogate is not novel, the LSTM surrogate model enabled the differentiable computing workflow and further saved an order of magnitude of computational time as compared to the VIC model running on CPU. Strikingly, using the same criterion (RMSE=0.05), it takes dPL roughly 25 minutes at 1/4² density on a single GPU (NVIDIA 1080Ti with 11GB memory) while it would take 33 days for a single CPU, or 475 minutes for a 100-core CPU cluster assuming perfect parallelism. Training the surrogate model (see Methods) also required multiple iterations of CONUS-scale forward simulations. All things considered, the new dPL framework brings a difference of 2-3 orders of magnitude in time, not to mention the savings in energy. This is despite that dPL trains large neural networks with thousands of weights. While there are more efficient variants of SCE-UA, we compared dPL to the standard algorithm because it is well understood and benchmarked, and its variants do not differ by orders-of-magnitude in efficiency.” {173-183}

New Figure 2. (a) Objective function (root-mean-square error, RMSE) for the testing period vs. number of forward runs per gridcell. Here a forward run for dPL means running through the same number of days as the specified training period, e.g., 30 random instances of 240-day-long simulations in a minibatch would be counted as ~ 20 runs for a 2-year training period. Dashed lines are for SCE-UA and solid lines are for *dPL*. The models were trained with 1 year’s worth of data. s16, s8, and s4 denote models trained with sampling densities of $1/16^2$, $1/8^2$, and $1/4^2$, respectively, where $1/16^2$ represents sampling one gridcell from each 16x16 patch. (b) Same as (a) but for models trained with 2 years’ worth of data. (c) RMSE vs computational time per gridcell. (d) Same as (c) for models with 2-year training data.

Minor Comments

Line 90: “a type of neural network with large depth” is an overly vague definition of deep learning. Please clarify that deep learning includes multiple layers that are often specialized to find patterns in the spatial and/or temporal structure of data.

Good point. We changed it to “*Recently, deep learning (DL)^{22,23}, neural networks with multiple layers and specialized architecture to learn patterns from spatial or temporal structure of the data.*” {Line 71-72}

Line 180: it would be helpful to mention the temporal length of the simulation up front and not just in the methods, where I believe 240 days was mentioned. Since this journal has a broad audience, the reader’s experience with a single run of a simulation could vary greatly depending on their field of expertise.

During a forward run, the code can run as long as there is input data. During training, there are two different concepts: instance length and the training period.

The instance length is what is used for one instance of an LSTM in a minibatch to go through at once during training, while the training period is what is available for LSTM to use as training data. Our Methods explained it as this (*bolded sentence is now added during revision*): “*We used a batch size of 300 instances and the length of the training instances was 240 days. Given one year or two years of training data, the code randomly selected grid cells and time periods with a length of 240 days within the training dataset to form a minibatch for training (**In contrast, SCE-UA calibration uses all available years of training data at once, as this is the standard approach**).*” {line 636}

Lines 191-192: what kind of CPU and GPU were used for the benchmarking? Please specify in the methods.

Please see our response to the same question above.

Figure 2: Can you place the matching marker next to each time series plot? Can you also use lines with greater contrast than blue and black? It is hard to see the differences between the two models.

Good suggestion. The figure is revised as below.

Figure 4: does infiltr have units? Also, it would be more readable to use infiltration in the title rather than INFILT. Why is the parameter range for SCE so much larger than for dPL?

INFILT is dimensionless and it denotes variable infiltration curve parameter of VIC (unitless). It is so much exactly because SCE-UA is ill constrained. Wildly different parameter sets can give similar results. Hence very random values can be found for a parameter like INFILT with SCE-UA. As discussed in

the paper, this problem is called parameter non-uniqueness or equifinality, and is a perennial problem in hydrology. Regionalizational method can partially overcome this issue, but they face many limitations of their own. dPL brought in much better constraint and optimization capability.

Changed as suggested.

Lines 320-331: why use NSE for one experiment and KGE for the other? It would be clearer to pick 1 and stick with it.

The main reason was to compare with the literature. For example, Beck20 did not provide NSE for comparison while the CAMELS study did not provide KGE.

We added this explanation to the Figure caption *“We used NSE for the CAMELS case and KGE for the global catchment case because these metrics were used by different papers and the main purpose of the case studies were to compare with the literature.”*

Figure 5: I am not sure how to interpret this plot? What should the CDF look like ideally? Is the green line a baseline or what you are trying to aim for?

It will be a little hard to explain the ideal line in words, but basically, the higher the values, the better, so the ideal CDF would be a curve that is a vertical line toward the right at NSE=1, but this never happens, and it’s difficult to visualize, so we added the following explanation *“For both panels, better models have curves to the right of the worse models.”*

Figure 6: it looks like another significant figure is needed for each label on the green axis.

That is correct. We fixed this. Thanks.

Line 617: how many models were trained as part of the tuning process? How much computational cost did this add?

To tune the hyperparameters, we ran 5 jobs for tuning batch size ; 15 jobs for tuning parameters of the AdaDelta optimization algorithm (3 parameters in AdaDelta algorithm); 3 jobs for s16 hidden size, 4 jobs for s8 hidden size, and 4 jobs for s4 hidden size. Tuning batch size and tuning parameters of optimization algorithm are based on s16 data density with 1-year training, about 5 sec. for each epoch running, 500 epoch for each model → about 40 mins.

Tuning of hyperparameters is often done for deep learning work. Some iterations are required to roughly tune the hyperparameter. We did not seek to find the best hyperparameters and took a somewhat *lazy* approach as we did not want to overtune them. Once they were in a reasonable range we stopped tuning them and simply used the same hyperparameters.

Another point to raise is that evolutionary algorithms like SCEUA also have hyperparameters. We had to tune ‘the number of complexes (sub-populations)’, ‘the number of members in a complex’ , ‘the number of members in a simplex’, ‘the number of evolution steps for each complex before shuffling’, and ‘the total number of points in an iteration’. Although there are some suggested values for these SCEUA hyperparameters, these hyperparameters will be tuned for our example. A similar *lazy* approach was taken for SCE-UA.

Line 620: do all the networks only use 1 hidden layer with LSTM?

LSTM does not have just one hidden layer. It has multiple gates that serve as hidden units, each of which is one fully-connected layer. These are explained in the equations.

Line 624: how many batches per epoch?

As answered earlier, in the revised manuscript, we removed the concept of epoch, but using “number of forward runs per gridcell” to quantify computation. This is because our epoch was defined in a somewhat unique way (a epoch corresponds to the probability that 99% of the time periods of all basins are used in the epoch) and frankly there is no algorithmic significance between epochs. Minibatch is the unit at which we update the weights.

Just for the curious,

In s16 density, 2-year training: 12 batches per epoch

In s8 density, 2-year training: 54 batches per epoch

In s4 density, 2-year training: 221 batches per epoch

Line 766: should specify that y is the prediction.

Modified as suggested: “ is the average modeled (predicted) value of all pixels”

REVIEWERS' COMMENTS

Reviewer #1 (Remarks to the Author):

The authors have done an excellent job revising their manuscript to address the constructive review comments. In my opinion, the paper can not be accepted in its current form.

Reviewer #2 (Remarks to the Author):

Thanks for considering my review comments. The revised manuscript is suitable for publication.
Hoshin Gupta.

Reviewer #3 (Remarks to the Author):

The major revisions the authors have made to the paper have greatly improved the presentation of the results. I do not have any further objections and recommend acceptance.